# Nanofluidic voidless electrode for electrochemical capacitance enhancement in gel electrolyte

Kefeng Xiao [1], Taimin Yang [2], Jiaxing Liang[1], Aditya Rawal[3], Huabo Liu[1], Ruopian Fang[1], Rose Amal [1], Hongyi Xu [2✉] & Da-Wei Wang [1✉]

Porous electrodes with extraordinary capacitances in liquid electrolytes are oftentimes incompetent when gel electrolyte is applied because of the escalating ion diffusion limitations brought by the difficulties of infilling the pores of electrode with gels. As a result, porous electrodes usually exhibit lower capacitance in gel electrolytes than that in liquid electrolytes. Benefiting from the swift ion transport in intrinsic hydrated nanochannels, the electrochemical capacitance of the nanofluidic voidless electrode (5.56% porosity) is nearly equal in gel and liquid electrolytes with a difference of ~1.8%. In gel electrolyte, the areal capacitance reaches $8.94\,F\,cm^{-2}$ with a gravimetric capacitance of $178.8\,F\,g^{-1}$ and a volumetric capacitance of $321.8\,F\,cm^{-3}$. The findings are valuable to solid-state electrochemical energy storage technologies that require high-efficiency charge transport.

[1] School of Chemical Engineering, The University of New South Wales, Sydney, NSW, Australia. [2] Department of Materials and Environmental Chemistry, Stockholm University, Stockholm, Sweden. [3] Nuclear Magnetic Resonance Facility, Mark Wainwright Analytical Center, The University of New South Wales, Sydney, NSW, Australia. ✉email: hongyi.xu@mmk.su.se; da-wei.wang@unsw.edu.au

The demands for safe and fast responding solid-state energy sources continue to rise. With short seconds-to-minutes charging time and long lifespan, supercapacitors (SCs) are in principle superior to batteries for high-power energy storage applications. Solid-state SCs have emerged as high-priority energy storage solution for on-chip and flexible electronic devices[1,2]. Solid-sate SCs with gel polymer electrolytes impart promising advantages over the traditional SCs with liquid electrolytes, such as easy maintenance, better reliability, and improved manufacture flexibility. However, as the developing electronics go more miniature, there is a growing technological desire for electrode materials delivering both high areal and volumetric capacitance, without compromising the gravimetric capacitance.

Porous electrodes are widely used in solid-state SCs. In general, embedding uniform ion penetration network in porous electrodes is key to achieving high capacitance in gel electrolytes. However, gel electrolytes are oftentimes poorly penetrative due to the entanglement and stickiness of cross-linked polymer chains. Such gel penetrability dependence of electrode capacitance causes a significant performance drop compared to electrodes in liquid electrolytes. Many reported works chose to alleviate the low gel penetrability by limiting the mass loading and thickness of electrodes ($<1$ mg cm$^{-2}$), which results in small areal capacitance. Enhancing gel electrolyte penetration through increasing the proportion of macropores ($>100$ nm) in the electrode is commonly applied (some cases also use less viscous sol electrolytes)[3–11], but this approach would impair the volumetric capacitance. Additionally, it has been stressed that the practical gravimetric capacitance of the electrode would drastically diminish when the mass of electrode-loaded electrolyte is counted[12–14]. These facts suggest that enhancing the electrode capacitance without compromising one other metric is usually constrained for solid-state SCs.

We propose a different electrode design concept for solid-state SCs with the purpose of achieving universal high capacitance on all metrics (gravimetric, areal, and volumetric). Our strategy is to use nonporous two-dimensional (2D) nanofluidic structure that is intrinsically dual conductive to electrons and ions as the electrode active materials in solid-state SCs. We validate the strategy using tungstate anion-linked polyaniline (TALP), a layered 2D conductive polymer-oxyanion structure[15,16]. In this work, we observed the laterally confined water in the layered TALP ($c = 1.18$ nm), which forms intrinsically hydrated nanofluidic channels that are inherently ionic conductive. The scale of the confined hydrated ion channel in TALP is close to double Debye length ($2\lambda$)[17,18], a parameter that is experimentally examined useful for high-density charge storage[10,12,19–21]. We also showed the robustness of the layered nanofluidic channels of TALP in keeping regular ion pathways under mechanical compaction, which is distinct from the deformable complex carbon networks[22,23]. This property allowed compressing powdery TALP particles to compact pellet electrode with a large apparent density of $1.8$ g cm$^{-3}$ and a very low porosity of 5.56%. The primary TALP particles sheared and fused in the pellets, which produced a spreading nanofluidic ion penetration network throughout the electrode even without external electrolyte flooding. The nanofluidic TALP pellet electrode showed almost equal high areal capacitances in liquid and gel electrolytes ($9.10$ vs. $8.94$ F cm$^{-2}$), as well as gravimetric and volumetric capacitance, which enables exceptional holistic capacitances that largely outstrip the state-of-the-art porous electrodes in solid-state SCs.

## Results

**Intrinsic nanofluidic ion channels**. The self-assembly of the layered structure of TALP is directed by the cooperating oxidative polymerization and hydrogen bonding interactions of monomeric aniline and oxotungstate via a one-pot process in an aqueous medium[15,16]. The unexfoliated original TALP particles consist of stacked nanosheets with high regularity, as demonstrated through transmission electron microscopy (TEM) analyses (Fig. 1a, b). Such unidirectional arrays of self-aligned nanochannels that are in atomic proximity distinguish the corrugated texture or hierarchical connectivity of the ion channels in restacked or crosslinked porous nanosheets[10,12,20,21,24,25]. The 1.18 nm lamellar periodicity of nanochannels in TALP, according to X-ray diffraction (XRD) measurement[15,16], is close to $2\lambda$ and is promising to store more charges and induce high ionic flux during charging[26]. Ultrasonic agitating the intact as-synthesized TALP particles in water or ethanol can extensively exfoliate the particles into ultrathin monolayer and few-layer nanosheets (Supplementary Fig. 1). Some strip-like cavities on the substrate nanosheets derive depth around 1.3 nm near to the monolayer thickness. The fine lattice structure of TALP nanosheet is determined through structure reconstruction of TALP particle basing on the reciprocal lattice (Fig. 1c). The lattice parameters of the averaged in-plane structure are determined to be $a = 6.86$ Å, $b = 7.60$ Å. The flake morphology of exfoliated nanosheets also implicates that the interlayer nanochannels of TALP are two-dimensional.

The hygroscopic oxotungstate species are apt to bind with water amid the aqueous-phase synthesis and endow TALP with hydrophilicity, demonstrating a water contact angle at 35° (Supplementary Fig. 2a). This material is also lipophilic with a hexane contact angle at 7° because of the aromaticity of polyaniline (Supplementary Fig. 2b). The amphiphilicity of this material promises it to possess good compatibility with a variety of electrochemical environments. The deuteration of interlayer water occupied inside the 2D nanochannels of TALP is achieved by grinding and soaking the particles in $D_2O$ and thorough drying under vacuum. The water dynamics is probed by 1D $^2H$ spectroscopy collected with solid-state NMR (ssNMR). This analysis identifies three modes of free (3 kHz), intermediate (33 kHz), and surface-bound (135 kHz) interlayer $D_2O$ in the deuterated particles (Fig. 1d)[27,28]. The bound $D_2O$ (135 kHz) is static and has hydrogen bonding with the functional groups on the channel surface[29]. The free and intermediate $D_2O$ between the two layers of surface-bound water are more mobile. This result indicates a three-layer model, which corresponds with a thickness of $\sim1$ nm[18]. Therefore, the TALP with sub-2nm nanochannels and conductive nanosheets resembles a typical 2D mixed ion-electron conductor, where the width of the ion transport path is on the $2\lambda$ length scale that is essential to attain high density of charge storage (Fig. 1e). The co-assembled nanosheets and nanochannels are stable due to the non-covalent cohesive hydrogen bonding, ionic interaction, and van der Waals forces. The synthetic water interlayers as intrinsic 2D nanofluidic channels are sufficient for the realization of swift ion transport through the interior of TALP particles with easy access to an exceptionally large internal electroactive interface. This inherent advantage of TALP circumvents the inevitable prohibition of ion accessibility across long diffusion distance in porous materials that leads to a substantial fraction of unusable space and low charge storage density.

**Uniform spreading ion penetration network**. TALP powder was compacted into a round pellet in a cylindrical mold (Fig. 2a and Supplementary Fig. 3a−c). Most TALP particles show intimate surfaces implicating the tight stacking between opposite nanosheets (Supplementary Fig. 3b), which is distinct from the micro-corrugated nanosheets that are prone to separate due to intercalation and exfoliation. The SEM image of the cross-sectional

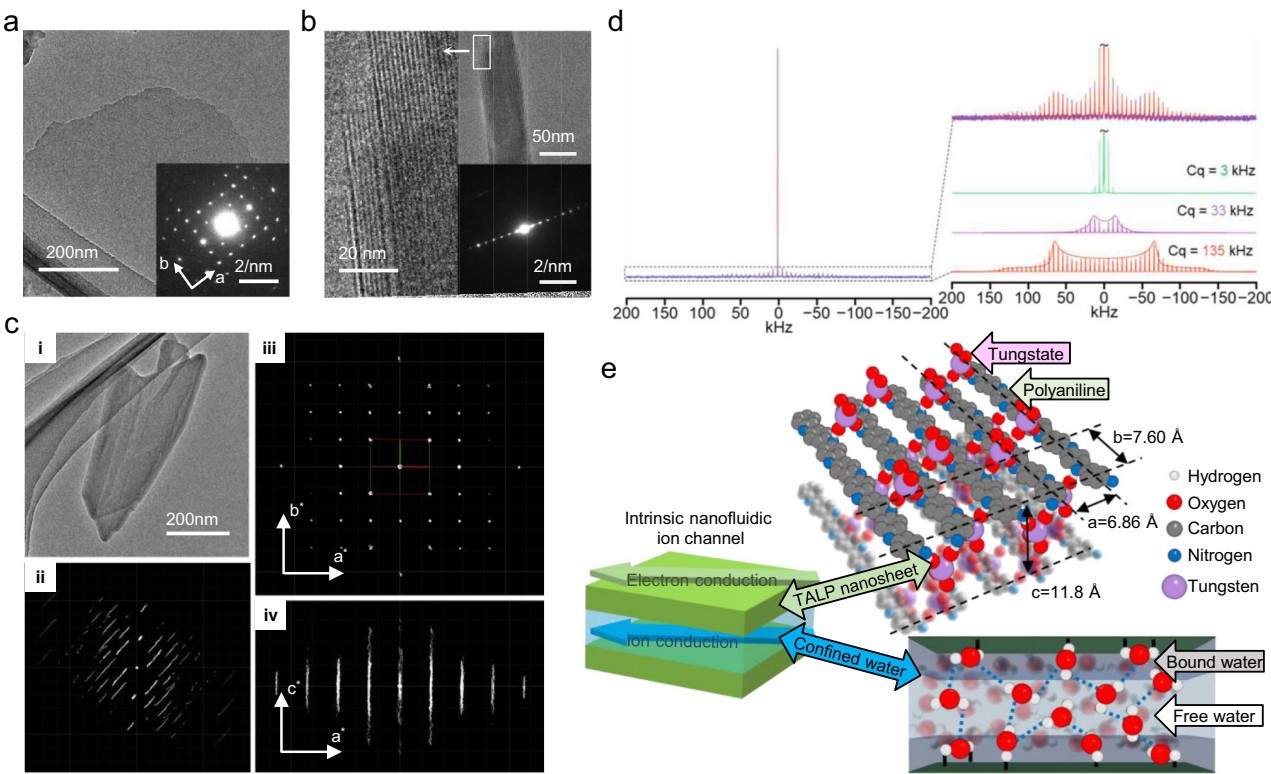

**Fig. 1 Intrinsic nanofluidic ion channels in TALP. a** The top-view TEM image of a delaminated TALP monolayer. The inset of **a** shows the electron diffraction pattern of the lateral structure. **b** The side-view TEM image of an unexfoliated TALP nanoparticle showing the layered structure. The inset of **b** shows the electron diffraction pattern of the ordered stacking. **c** Reconstruction of the 3D structure of TALP particle through the reciprocal lattice. **c**-i TEM image of a typical delaminated TALP few-layer nanosheets. **c**-ii Reconstructed 3D reciprocal lattice of the thin plate-like particle. **c**-iii and **c**-iv Two 2D slices, (hk0) and (h0l) cut from the 3D reciprocal lattice. The unit cell axes $a^\star$ and $b^\star$ are shown in red and green, respectively. The discrete diffraction spots in (**c**-iii) suggest the arrangement of TALP nanosheet is highly ordered. The streaky diffraction spots in **c**-iv indicate the few layer nanosheets are very thin. **d** The 1D $^2H$ ssNMR spectrum of $D_2O$-TALP and the simulated $^2H$ quadrupolar results revealing the ionically conductive nature of confined water in the interlayer space of TALP. $C_q$: quadrupolar coupling constant. **e** A structure model of the intrinsic nanofluidic ion channels in TALP particle including the illustrative structure of TALP nanosheet and the confined water layer in the interlayer space.

surface of TALP pellet shows that the voidless interior texture is caused by the particle deformation and fusion under compression (Fig. 2b). The deformation of TALP particle is plausibly correlated with the lubricity of nanosheets led by non-covalently bounded dynamic water molecules contained in the interlayer space[21,30]. In general, for the ordinary porous materials like graphene, the retention of mediocre porosity—such as >40% in three-dimensional graphene networks[24,31]—is regarded as an essential characteristic of electrodes capable of rapid charge delivery, where the thorough electrolyte infilling of pores is critical[22]. The micro-CT analysis unveils the dense interior of TALP pellets with calculated porosity of 5.56% that is way lower than that of the porous materials (Fig. 2c). The surface area for the pellet is about $0.4 \, m^2 \, g^{-1}$, reduced by 98% compared to the $22 \, m^2 \, g^{-1}$ for the TALP powders, which indicates the significant loss of inter-particle voids and external surfaces after compaction. The negligible pore volume of $0.04 \, cm^3 \, g^{-1}$ for TALP pellet is coherent with its superdense interior texture. Furthermore, the 2D nanofluidic ion channels are preserved after compression and the increasing compaction pressure imposes negligible destructive effect to the regular stacking of the nanochannels (Fig. 2d). The above features suggest that the TALP pellet contains tightly fused nanofluidic particles that are capable of uninterrupted ion transport in bulk disregard its ultralow porosity.

XRD results of the standing and lying TALP pellets reveal the intensity difference of the characteristic peak for layered stacking (Fig. 2e). The noticeable peak intensities in both the standing and

lying samples imply the bi-directional orientation of nanosheets along the pellet axial and radial directions, despite the preference along the axis (Supplementary Fig. 4a). The total bright areas on the cross-sectional surface of the TALP pellet are more than five times that on the external top surface, based on the statistical brightness analysis of the scanning electron microscopy (SEM) images (Fig. 2f). The bright zones in the SEM images are regarded as the basal planes of stacked TALP nanosheet because of their high nanoscale uniformity and the tendency of surface charging (Supplementary Fig. 4b). Accordingly, the dark zones represent the edge planes of the 2D frameworks due to their fast charge dissipation. The portion of bright zones on the top surface is ~10% but increases to ~65% on the cross-sectional surface. Additional investigation on the lubricating and sliding behavior of TALP particles should be conducted to understand the orientation upon compression.

The high-density fusion of TALP particles suggests the connectivity of the nanofluidic channels in the pellet electrode. To probe the bulk ion conduction of the TALP electrode in solid-state SCs, the electrochemical impedance spectroscopy (EIS) analysis was conducted by using symmetric two-electrode cells with leak-free solid gel electrolyte membrane (Supplementary Fig. 5a, b). In order to ensure a credible evaluation of the nanofluidic ion conduction at solid-state, we avoided soaking the TALP electrode with any forms of electrolyte before the cell was assembled; however, electrolyte pre-infilling is a common treatment of porous electrodes in solid-state SCs[3–11].

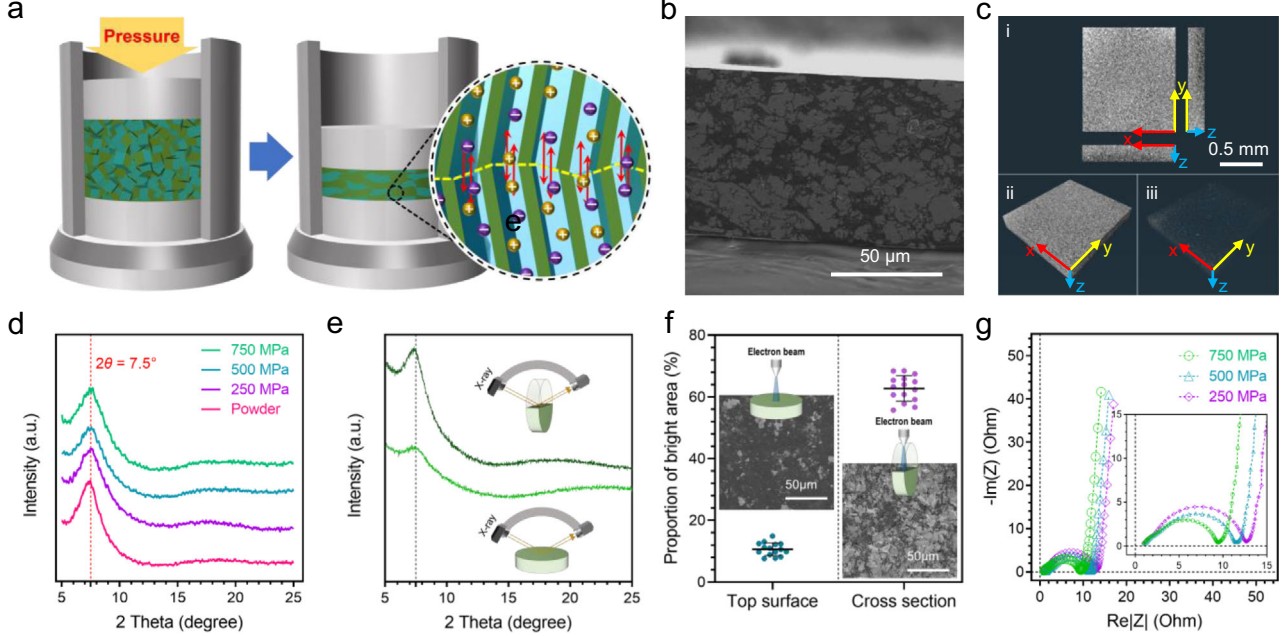

**Fig. 2 The uniformly distributed ion penetration network throughout the pellet. a** A schematic illustration of the mechanical compaction of TALP particles with the resulting inter-particle connectivity of nanofluidic ion channels. **b** The SEM image of the cross-sectional surface of a TALP pellet. **c** Micro-CT images of TALP pellet. **c**-i Top view and cross-section view. **c**-ii The reconstructed micro-CT image. **c**-iii The pore structure. **d** The XRD patterns of TALP powders and pellets fabricated under different pressures, which show the consistent structural features of layered nanofluidic channels. **e** The XRD profile collected from the standing and lying TALP pellets. The intensity change of X-ray diffraction is a statistical indicator of the alignment direction of the basal planes of 2D ion channels against the incident X-ray. **f** The statistical brightness analysis of the SEM images collected from the top surface and cross-sectional surface of the TALP pellets. **g** Nyquist plots of TALP electrode fabricated at different pressures with 5 mg cm$^{-2}$ mass loading. The measurement was conducted in a symmetric solid-state EC cell with a leak-free gel electrolyte.

The complex−plane impedance plots (Nyquist plots) display the characteristic semicircles at high-medium frequency and straight tails at medium-low frequency (Fig. 2g). As the compression pressure increases, the charge transfer impedance steadily decreases as a result of the reducing inter-particle resistance. The knee frequency represents the beginning point where the ion transport dominates the electrode process. In the Nyquist plots in Fig. 2g, when the frequency is lower than the knee frequency, the tails stay nearly vertically confirming a fast kinetics of ion transport. The relationship between the knee frequency and diffusion coefficient can be described as following[32,33]:

$$D = \omega_0 \times L^2 \qquad (1)$$

Where, $D$ is the diffusion coefficient; $\omega_0$ is the knee frequency; $L$ is the thickness of the electrodes. The values of both knee frequency (~2.7 Hz, Supplementary Fig. 6) and time constants (~0.37 s) are coherent at all pressures. The nanofluidic ion diffusion coefficient for TALP electrode is about $7.19 \times 10^{-9}$ m$^2$ s$^{-1}$ that is two-to-three orders of magnitude higher than that of solid gel electrolytes[34]. This fact remarks that the ion transport in TALP-based solid-state ECs is more likely limited by the solid gel electrolyte[4,7].

**Electrochemical behaviors and capacitive performance**. The capacitive behaviors of TALP electrode in liquid and gel electrolytes are compared by using cyclic voltammetry (CV) (Fig. 3a). For liquid-state SCs, the TALP electrodes are flooded using liquid electrolyte to establish a proper baseline for comparison (Supplementary Fig. 5c). The CV curve of TALP symmetric cell in the liquid electrolyte can be seen as a combination of the CV curves of positive and negative electrodes (Supplementary Fig. 7). Such CV curve indicates that TALP electrode couples the surface-controlled faradaic and non-faradaic charge storage

mechanisms[16]. The CV curve of TALP in gel electrolyte basically replicates that in a liquid electrolyte, suggesting the identical capacitive behaviors of TALP in both electrolytes.

The slope of the low-frequency part of the Nyquist plot of TALP electrode in liquid and gel electrolyte are almost the same (Fig. 3b), which indicates the ion mobility in the electrode is independent from the type of electrolyte. This phenomenon implicates the capacity of TALP for solid-state ionic conduction that is brought by the intrinsic nanofluidic channels. The activation energy for nanofluidic ion transport in the TALP electrode is 101.2 meV derived from temperature-dependent EIS (Fig. 3c and Supplementary Fig. 8), using the following equation[35,36].

$$\sigma T = A \exp(-E_a / k_B T) \qquad (2)$$

where, $A$ is the pre-exponential factor, $E_a$ is the apparent activation energy, $k_B$ is the Boltzmann constant, $\sigma$ is the ion conductivity, and $T$ is the absolute temperature. This value is much lower than that of proton diffusion in nanofluidic channel built from exfoliated vermiculite (190 meV)[37]. The stability of TALP electrode for repeatedly charging and discharging the nanofluidic channels is examined by 88.6% retention after 5000 cycles at 20 mA cm$^{-2}$ (Supplementary Fig. 9).

The charge−discharge curves of TALP electrode (10 mg cm$^{-2}$) in a symmetric cell with gel and liquid electrolyte are demonstrated in Fig. 3d. The TALP electrode exhibits close specific capacitance in a gel electrolyte to that in liquid electrolyte. The capacitances of TALP electrode and porous graphene electrode in gel and liquid electrolyte at different rates are demonstrated in Fig. 3e. The TALP electrode exhibits a specific capacitance over 170 F g$^{-1}$ at 1 mA cm$^{-2}$ with a negligible capacitance gap between gel and liquid electrolytes at various current densities. In contrast, when using gel electrolyte, the graphene foam electrode exhibits very low

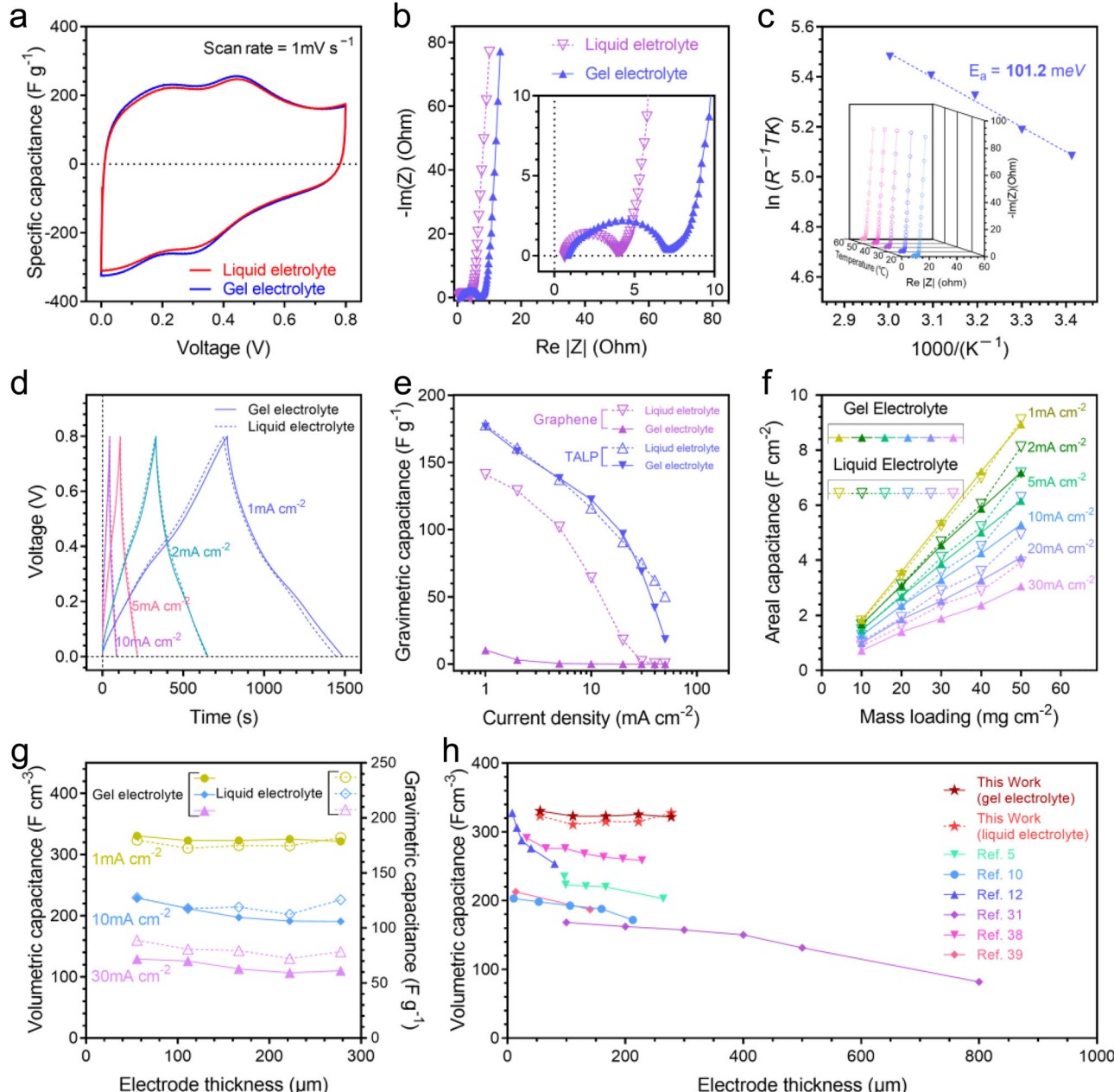

**Fig. 3 Electrochemical behaviors and capacitive performance of TALP electrode. a** CV profiles for TALP electrode in liquid and gel electrolytes. **b** Nyquist plots of TALP electrode with 10 mg cm$^{-2}$ mass loading in liquid and gel electrolytes. **c** The Arrhenius plot for the TALP electrode demonstrates the activation energy of ion transport in the nanofluidic ion network in the TALP electrode. The inset is the Nyquist plots of TALP electrode at different temperatures. **d** Galvanostatic charge–discharge profiles for TALP electrode (10 mg cm$^{-2}$) in liquid and gel electrolytes. **e** The capacitance comparison among TALP electrode in different electrolytes and that of the graphene foam electrode. When gel electrolyte was used, the TALP and graphene foam were not soaked with any form of electrolyte. **f** The areal capacitance of TALP electrodes (mass loading ranging from 10 to 50 mg cm$^{-2}$) versus current densities (1–30 mA cm$^{-2}$) in gel and liquid electrolytes. **g** The relationship between electrode thickness and volumetric capacitance of TALP electrodes at different current densities. **h** The volumetric capacitance versus electrode thickness in liquid and gel electrolytes[5,10,12,31,38,39].

specific capacitance of 10.6 F g$^{-1}$ at 1 mA cm$^{-2}$, which is 1/15 of that in liquid electrolyte. The stark contrast in the capacitance gap in liquid and gel electrolytes remarks the advantage of the nanofluidic channels of TALP for solid-state SCs, even without gel permeation.

The approximate weight of passive components in commercial cells is suggested around 10−30 milligrams per square centimeter[14,24]. Therefore, when electrodes have mass loading of 10 mg cm$^{-2}$, the device performance is often less than 50% of that of the electrode[14,24]. Acquiring high capacitance at high

levels of mass loading is essential to minimize the overhead for achieving superior device performance. A series of compact TALP electrodes that has mass loading from 10 to 50 mg cm$^{-2}$ were made under 750 MPa pressure, with corresponding electrode thickness from 56 to 278 μm. The multiplying relation between volumetric capacitance (F cm$^{-3}$) and electrode thickness (μm) produces areal capacitance (F cm$^{-2}$). For this reason, increasing areal capacitance simply relies on adding high mass loading or thickness of the electrode. However, the areal capacitance does not always linearly increase with the increase of electrode mass

loading due to the accumulating thickness that imposes difficulty for electrolyte penetration, which is common in practice and especially for the electrode using gel electrolyte[31]. As a result, the capacitance of porous electrode will level off towards a maximum value[4,7]. Due to the electrolyte-independent capacity of ion transport, such maximum value for TALP is theoretically higher than that of the porous electrodes. Within the mass loading ranging up to 50 mg cm$^{-2}$ (278 μm), TALP electrode keeps the linear relationship between the areal capacitance and electrode mass loading and achieves an ultrahigh areal capacitance of 8.94 (solid line) and 9.10 (dash line) F cm$^{-2}$ when using gel and liquid electrolyte, respectively (Fig. 3f). Figure 3g shows the relationship between volumetric capacitance and thickness of TALP electrode. It reveals that the volumetric capacitance of TALP electrode (330.7−321.7 F cm$^{-3}$ for liquid electrolyte and 337.6−310.5 F cm$^{-3}$ for gel electrolyte) scarcely decreases with the increasing electrode thickness under a low current density of 1 mA cm$^{-2}$. Under a high current density of 30 mA cm$^{-2}$, the decrease of specific capacitance is less than 20% in which a considerable portion is brought by the resistance of gel electrolyte. Such thickness-independent capacitive performance of TALP can be attributed to the percolating nanofluidic ion channels. By comparison, some porous electrodes usually demonstrate gradual loss of volumetric capacitance as the electrode thickness (mass loading) increases[5,10,12,31,38,39] (Fig. 3h and Supplementary Fig. 10).

**Comparative advantages of nanofluidic electrode**. When the holistic capacitance metrics are considered (Fig. 4a), nanofluidic TALP electrodes are more appropriately positioned for compact solid-state SCs than gel-infilled porous electrodes[5–9,38,40,41]. The comparison of holistic charge metrics (Supplementary Fig. 11) also implicates the similar trend. The TALP electrodes deliver much higher volumetric capacitance (322–330 F cm$^{-3}$) than that of the typical porous electrodes (21.7–230 F cm$^{-3}$). Electrode with high areal capacitance, yet small thickness, is desirable for compact solid-state SCs. With similar areal capacitance (~1.8 F cm$^{-2}$), TALP electrode attains a thickness of only 56 μm, which is nearly half of the composite film made of graphene/conductive polymer (~96 μm, ~2.2 F cm$^{-2}$)[5] or about six times less than the carbon cloth (~340 μm, ~1.8 F cm$^{-2}$)[8]. Many gel-infilled porous electrodes have a low volumetric capacitance that restricts the areal capacitance (0.174–3.38 F cm$^{-2}$) despite the large thickness

(up to 1500 μm)[6]. TALP electrode allows extra-large areal capacitances (8.94 F cm$^{-2}$) in nonporous and relatively thin (278 μm) electrodes because of its nanofluidic ion diffusion network that is uniformly distributed throughout the whole electrode. For solid-state SCs, porous electrodes hardly deliver compact capacitive performance comparable to that of the nanofluidic TALP electrode.

Preserving high gravimetric capacitance at large mass loading is the key to high areal capacitance. The TALP electrodes can acquire high gravimetric capacitance about 180 F g$^{-1}$ at exceptionally large mass loading of 50 mg cm$^{-2}$, which is on par with those recorded at low mass loading levels (~10 mg cm$^{-2}$) and exceeds those at relatively high mass loading (~20 mg cm$^{-2}$). When the practical gravimetric capacitance is considered, the total mass of gel-infilled porous electrodes needs to include the gels occupied by the pores. Therefore, the values of gravimetric capacitance in Fig. 4a will be scaled down to those compared in Fig. 4b. This translational capacitance variation can be normalized using a performance factor, which indicates the influence of gel infiltration on the usable gravimetric capacitance rounded to the total mass taken up by the electrode volume. The performance factor decreases monotonously as gel-permeable porosity increases. Being ion conductive but gel impermeable, the nanofluidic TALP electrode is promising to uplift the overall gravimetric performance of solid-state SCs.

## Discussion
The practical significance of percolating nanofluidic channels for solid-state SCs is detailed in this work. TALP electrode with interconnecting intrinsic nanofluidic ionic paths demonstrates great potential for large areal capacitance with both gel and liquid electrolytes (8.94 and 9.10 F cm$^{-2}$). In contrast to the porous electrodes that are restrained by the electrolyte-dependent capability of charge storage, the electrolyte-independent capacitive behavior of the intrinsic nanofluidic TALP electrodes results in a significantly higher areal performance, as well as an extraordinary holistic performance. The uniformly distributed ion paths are critical for achieving the remarkable volumetric (>320 F cm$^{-3}$) and gravimetric (>170 F g$^{-1}$) capacitances over a wide range of electrode thickness up to 278 μm and high mass loading up to 50 mg cm$^{-2}$ in gel electrolyte. Adjustment of the channel size will promisingly extend its use to gel electrolytes based on organic solvents or ionic liquids. This progress demonstrates the

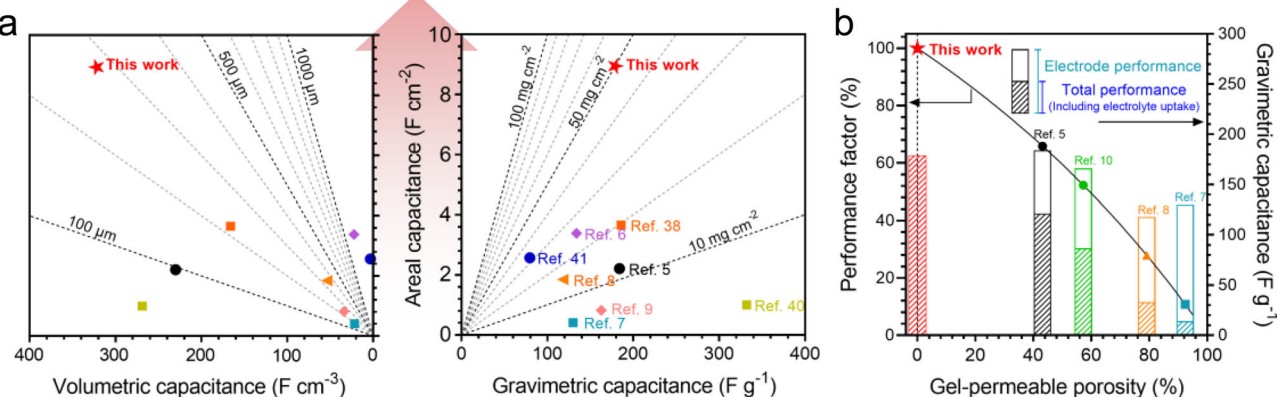

**Fig. 4 Benchmarking of nanofluidic electrode in gel electrolyte. a** The holistic comparison of areal capacitance versus volumetric and gravimetric capacitances between nanofluidic electrode (this work) in gel electrolyte and gel-infilled porous electrodes[5–9,38,40,41] (Supplementary Table 1). **b** Projection of the gravimetric capacitance of electrode in (**a**) when the mass of infilled gel electrolytes is included to account for the total mass accommodated by the volume of electrode. The resource data and calculation methods are detailed in Supplementary Tables 2 and 3. The theoretical density (1.5 g cm$^{-3}$) of a model gel electrolyte (H$_2$SO$_4$/PVA) was used in the translation[40].

advantages of 2D nanofluidic materials as electrode materials for solid-state SCs and promises a universal strategy for designing electrodes used in solid-state electrochemical energy storage devices.

## Methods

**Synthesis of TALP**. The TALP used in this study was synthesized following previous reports[15,16]. Typically, TALP synthesis was fulfilled through oxidation polymerization of aniline in presence of tungstate. With continuous stirring, 50 mL aqueous solution of ammonium meta-tungstate (0.1 M by W atom)/ammonium persulfate (0.3 M) was added dropwise to 50 mL aqueous solution of $H_2SO_4$ (0.1 M)/aniline (0.2 M). The mixing was fulfilled under 5 °C and the duration was approximately 30 min. The mixed solution was kept under 5 °C and continuously stirred for 24 h. The product was collected after filtration and washing and subject to drying in a vacuum at 60 °C for 24 h. All chemicals were purchased from Sigma-Aldrich with purity above 98%, and directly used without further purification.

**Fabrication of TALP pellet electrode**. The TALP powder was mixed and ground with carbon black at a weight ratio of 9:1. The binder-free electrode was made by cold pressing the dry mixture in a cylindrical die with a diameter of 10 mm. The reported capacitance values in this study are based on the total mass, area, or volume of the pellets containing 90% TALP and 10% carbon black. TALP pellet is fabricated from pure TALP powder, and its fabrication process is the same as that of TALP electrode. Note, TALP electrode is used for electrochemical tests and TALP pellet is used for all the other tests.

**Fabrication of PVA/PEG gel electrolyte**. 7.5 g polyvinyl alcohol (PVA, Mw. 80,000, Sigma-Aldrich) was dissolved in 100 mL deionized water at 90 °C to obtain a viscous clear solution. Then 7.5 g Polyethylene glycol (PEG, Mw. 3000, Sigma-Aldrich) and 1.5 g $Na_2SO_4$ (99.8%, Sigma-Aldrich) were added to the PVA solution at 90 °C under stirring until the solids were completely dissolved. The as-made sol was casted and dried on the glass to get the gel membrane. The peeled gel membrane was saturated in 1 M $Na_2SO_4$ aqueous solution. The gel membrane was then fully dried before use.

**Assembly of slid-state SC cells**. The model solid-state SC cells were made by sandwiching the gel membrane with two symmetric TALP electrodes or porous graphene electrodes with a mass loading of 6 mg $cm^{-2}$. Before assembly, the TALP and porous graphene electrodes were fully dried and not soaked with liquid or sol electrolyte.

**Assembly of liquid-state SC cells**. The flooded SC cells were made by sandwiching the separator between two symmetric TALP electrodes. Excessive aqueous 1 M $Na_2SO_4$ electrolyte was added to the cell container to ensure the electrodes and separators are fully soaked. Before adding electrolyte, the TALP electrodes were kept dry.

**Materials characterization**. SEM characterization was carried out on the FEI Nova NanoSEM 450 FESEM at 5 kV. XRD was conducted on a Bruker D8 Thin-Film XRD with Cu Kα radiation (λ = 1.54056 Å) with a step interval of 2° $min^{-1}$. Focused ion beam processing and image collection were performed with Carl Zeiss AURIGA® CrossBeam® Workstation. Atomic force microscopy results were collected from a Bruker Dimension ICON SPM with tapping mode. Theta optical tensiometer was used to measure the contact angle. Micromeritics Tristar 3030 was utilized for porosity analysis.

**TEM characterization**. Samples were crushed into fine powders in a mortar. Then the fine powders were transferred to lacy-carbon copper grids by directly spray them onto the grid. TEM images and diffraction patterns were taken by using JEOL JEM-2100 FEG TEM operated at 200 KV. A Gatan Ultrascan CCD camera was used for collecting HRTEM images and a Gatan Orius CCD detector was used for collecting diffraction patterns. The 3D ED data were collected on a JEOL JEM-2010 LaB6 TEM equipped with an Amsterdam Scientific Instrument Timepix detector using Instamatic software[42]. The rotation electron diffraction (RED) software suite was used to process 3D ED data[43]. Note that the yield of crystalline sample on TEM grid is low and this phenomenon is coherent with the quasi amorphous feature of XRD for TALP.

**Micro-CT analysis**. The micro-CT imaging was performed using the HeliScan micro-CT system with a GE Phoenix Nanofucus tube with a diamond window and a high-quality flatbed detector (3,072 × 3,072 pixels, 3.75 fps readout rate). The samples were scanned in a circular trajectory with the following setting: 80 kV, 85 μA (tube current), exposure time 0.47 s, 3 accumulations, 0.75 Al filter, and 2880 projections per revolution. The voxel size obtained from this sample is 1.6 μm. The tomographic reconstruction was performed using QMango software developed by

the Australian National University. The porosity is calculated from the ratio of pore volume and the total volume of the electrode analyzed.

**Solid-state nuclear magnetic resonance (ss-NMR)**. Deuterated TALP powders were made by soaking normal TALP in $D_2O$ (70% D, Cambridge Isotope Laboratories, Inc.) for 1 week to substitute the interlayer $H_2O$. Prior to ssNMR measurement, the deuterated POAFs were dried overnight under vacuum at 100 °C. The ssNMR was run on a Bruker Advance III 300 MHz spectrometer with a 7 Tesla superconducting magnet operating at frequencies of 300 MHz and 46 MHz for the [1]H and [2]H nuclei respectively. Approximately 80 mg of sample was packed in a 4 mm zirconia rotor fitted with a Kel-F cap and spun to 6 kHz at the magic angle in a 4 mm HX double-resonance MAS probe head. The [2]H NMR signal was acquired with a solid echo sequence using optimized hard 1 μs radiofrequency pulses, with 60 kHz [1]H SPINAL decoupling and 1 s recycle delays. A total of 30 k signal transients were co-added to yield sufficient signal to noise and the overall signal was simulated using the Dmfit software to extract the quadrupolar parameters.

**Electrochemical measurements**. Common electrochemical measurements were conducted in two-electrode liquid-state or solid-state cells. EIS at different temperatures, cyclic voltammetry, and galvanostatic charge/discharge were all conducted on a Biologic VSP potentiostat. Impedance was collected from 10 mHz to 200 kHz. The current density for the galvanostatic test ramped from 1 to 20 mA $cm^{-2}$. Cyclic voltammetry was run at 1 mV $s^{-1}$. The cell voltage was 0−0.8 V.

## Calculations

The capacitance of a single electrode in this study was calculated from the data acquired with symmetric two-electrode cells.

The areal capacitance ($C_a$) was calculated as following:

$$C_a = 2I_a \times t/V \qquad (3)$$

where, $I_a$ represents current density (A $cm^{-2}$), $t$ represents discharge period (s) and $V$ is the voltage window of discharge.

The gravimetric specific capacitance ($C_g$) was calculated as following:

$$C_g = \frac{C_a}{m_{electrode}} \qquad (4)$$

where, $m_{electrode}$ is the mass loading (g $cm^{-2}$).

The volumetric capacitance ($C_v$) was calculated as following:

$$C_v = \frac{C_a}{L} \qquad (5)$$

where, $L$ is the electrode thickness.

As shown in Fig. 3d, the curve of the voltage vs. charge/discharge time is slightly deviated from linearity, which indicates the capacitive behavior of TALP electrode is not ideal. The instantaneous capacitance ($C_{ins}$) of each segmental voltage windows (d$V$) is defined different from the average capacitance for the entire voltage window ($V$). Therefore, it is necessary to calculate the $C_{ins}$ of the differential voltage window using the following equation:

$$C_{ins} = \frac{I \times dt}{dV} \qquad (6)$$

Where, $C_{ins}$ represents the instantaneous capacitance of the differential voltage windows, $I$ represents the current density applied in GCD test, d$t$ represents the segmental discharge period for time $t$, d$V$ represents the segmental voltage window for time $t$. The $t−V$ or $Q−V$ curve for d$t$ or d$V$ period is believed linear. The average capacitance ($C_{ave}$) on the actual voltage window $V$ can be calculated by using the following equation:

$$C_{ave} = \frac{1}{V} \int C_{ins} dV \qquad (7)$$

Where, $C_{ave}$ represents the average capacitance of the whole voltage window, $V$ represents the whole voltage window, $C_{ins}$ represents the instantaneous capacitance of the segmental voltage windows, and d$V$ represents the segmental voltage window. The example calculation is shown in Supplementary Note, which

reveals that the average capacitance is very close to that calculated from $C = I \times t/V$ that is used in this work to ensure consistency with the literature.

## Data availability

The data that support the findings and conclusions of this study are published online in the Article and its Supplementary Information file. Additional information with relevance is available from the corresponding authors on reasonable request.

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

## Acknowledgements

This work was financially supported by the Australian Research Council Discovery Project (DP190101008), Future Fellowship (FT190100058), the UNSW Scientia Program and the Swedish Research Council (Starting Grant 2017-05333). The authors thank Dr Qiang Zhu for conducting the FIB analysis. The authors acknowledge the scientific and technical assistance from Mark Wainwright Analytical Center for the use of facilities at the Electron Microscope Unit at UNSW and the Solid Spectroscopic Facilities. The authors further acknowledge the Tyree X-ray CT Facility, a UNSW network lab funded by the UNSW Research Infrastructure Scheme, for the acquisition of the 3D μXCT images.

## Author contributions

D.-W.W. planned and supervised the research. K.X. conducted material synthesis, electrochemical measurement, SEM, AFM, and XRD. J.L. and A.R. conducted ssNMR analysis. J.L. collected contact angle data. H.L. performed pellet preparation and electrochemical analysis. T.Y. and H.X. carried out TEM analysis. D.-W.W., K.X., R.F., and R.A. analyzed the results. D.-W.W., K.X., H.X., A.R., R.F., and R.A. wrote the paper.

## Competing interests

All authors declare no competing interests.
