## [Peer review file · Nature Communications]

Reviewers' Comments:

Reviewer #1:

Remarks to the Author:

In this article, the authors demonstrate compact capacitive energy storage in a solid-state electrochemical capacitor with thick electrodes. The authors have achieved bidirectional dense alignment of sub-2 nm nanofluidic ion channels in compacted electrodes. The authors assign it to nanofluidic superionic conduction that is 2-4 orders of magnitude faster than in solid gel electrolytes. They used two 2D conductive and hydrophilic polymer-oxyanion frameworks (POAFs) to construct highly aligned ion channels because of their clay-like structure and rheology. This is an interesting idea, but I'm not convinced if the system under study lives to expectations. I cannot recommend the paper for publication, at least not in the current state.

While the obtained capacitance values are high, it's important to remember that the voltage window is narrow (0.8 V), the charge/discharge rate is very low (1 mV/s in CV) and the energy and power density values are going to be low as well. Have the authors calculated energy and power density for their devices, even those with relatively thin electrodes?

It's important to remember that supercapacitors are power devices. If the power is low, they have no advantage over batteries. However, CVs were recorded at 1 mV/s (800 min to charge = >13 min). GCD curves show charging in excess of an hour. These charging rates are too low, even for gel electrolyte supercaps. Even conventional activated carbon electrodes can operate at a millimeter thickness at those rates. How can this issue be addressed?

A 10% loss of performance after 4000 cycles is not very impressive either. It's actually quite typical for pseudocapacitive electrode to show about 10,000 cycles lifetime (till 80% of the initial capacitance value). What is the main mechanism of degradation?

The authors claim that the areal and volumetric capacitance can be, respectively, 50 and 15 times higher compared to conventional porous supercap electrodes. Coarse-grained porous carbon electrodes with thickness up to 0.8-1 mm have been reported and showed comparable areal capacitance at a significantly higher rate in organic electrolytes. Also, graphene and MXene were vertically aligned to produce thick electrodes operating at significantly higher rates in liquid and gel electrolytes. I recall seeing a paper from China reporting MXene-gel electrolyte devices with 10 F/cm² capacitance. Pumera's group published high values of capacitance for MXene-ionogel devices in ACS AMI recently. An appropriate comparison should be provided.

Why is the vertical alignment of lamellas was maintained after uniaxial pressing (schematic in Fig. 2a)? 2D flakes usually align in-plane, normal to the applied force, upon compression.

Some of the material characterization data, such as AFM analysis of flakes and photos of pellets should be moved to main text. This will help readers to understand the material under study. It's difficult to read and understand the article without having SI open simultaneously.

The Nyquist plots suggest that the ionic conductivity is good, but there are large semicircles. Can the charge transfer problem be resolved by adding materials with high electronic conductivity, such as 2D Ti₃C₂ or rGO?

The authors report that GCD data (Fig. S9) were used for capacitance calculations. They should be reported in the main manuscript, along with CVs and Nyquist plots. Due to nonlinear discharge curves and non-rectangular CVs, the authors need to integrate (see, e.g, T.S Mathis in Adv. En. Mater., 2019). The equation for areal capacitance shown on page 18 was derived for an ideal capacitor. Therefore, I recommend recalculating all capacitance values.

Reviewer #2:

Remarks to the Author:

The development of high volumetric and high areal energy densities has been an attractive objective for the current pursuit of solid-state supercapacitors, which can be promising for powering different miniaturized electronic components. However, the current strategies utilized for porous electrode materials have failed to provide high energy densities when solid-state electrolytes are employed, which are mostly due to the limited ion diffusion in solid-state devices. In this work, the authors present the superdense solid-state energy storage using aligned nanofluidic frameworks based on POAFs. They have achieved excellent solid-state capacitance per footprint area and volume, which are superior to the state-of-art porous electrode materials. Overall, the achieved results are interesting and can be potentially published in Nature Comm. However, before further consideration, the following critical issues need to be addressed:

-The highly aligned sub-2 nm pores with fast ion diffusion are the core concepts of the current work. However, there is little structural proof that can disclose the sub-2 nm hydrated nanofluidic channels based on the synthesized POAFs. Actually, the chemical nature and chemical descriptions of the key materials POAFs are poorly defined in the current work. It is really hard to understand what are the chemical structures of POAFs, and how the sheets are stacked in the solid-state through non-covalent interactions. From Fig 1 and 1b, it is also difficult to discern the interlayer structures of POAFs; therein, the authors shall provide high-resolution TEM images to reveal the interlayer and intralayer structures.

-The swift ion transport through the POAF frameworks is crucial in the current work, but why sub-2 nm nanochannels can be the best? More understanding and proof of ion transport by comparison of different samples can be helpful. It seems that with high-pressure compression, it is possible to tune the nanochannels' sizes. Did the authors check other channel sizes dependent ion transport and electrochemical performance?

Minor comments: the calculation of knee frequency and time constants shall be provided in SI. The equations for the analysis of ion diffusion coefficient and activation energy shall be included in the main text.

Reviewer #3:

Remarks to the Author:

The authors demonstrated the conductive polymer-oxyanion frameworks (POAFs) possessing dense alignment of sub-2nm ion channels for rapid ion diffusion. When fabricated in thick electrodes, POAFs exhibit excellent areal and volumetric capacitance in solid-state aqueous electrolyte. While this manuscript needs a major revision before its consideration for publication. The comments are presented as follows.

(1) It is suggested to introduce the research background of POAFs. The novelty of this manuscript should be further explained when considering their published articles related to POAFs (Ref. 23 and 24)

(2) In page 6-7, the authors claimed "the predominant orientation of nanosheets along the pellet axis" based on XRD and SEM analyses. It is better to explain why the POAFs nanosheets are selectively orientated solely upon compaction. Is there something like inductive effect?

(3) The comparison of capacitance performance of this work with others lacks rigor (Fig. 3f, 4a). For example, Ref 10, the electrodes were measured in IL electrolyte, different from this work using aqueous electrolyte. As we known, IL electrolyte gives lower capacitance but significantly wider potential window. In addition, state-of-the-art pseudocapacitive examples should be included for comparison.

(4) The specific surface area and pore size distribution of POAFs should be provided.

(5) The thickness values of compacted POAFs pellets with different mass loadings should be provided.

Authors' response: These comments are valuable. We have carefully reinvestigated the data and conducted more control experiments. Based on the new findings, we have made major changes in the Revision and these are summarized here.

We revisited the claim of 'highly aligned ion channels'. We now consider it a less significant and unproven factor for the electrode behaviour in solid state gel electrolyte. Our rationales are based on two reasons. Firstly, we conducted a comparative test on porous graphene electrode (made from thermal reduction of reduced graphene oxide) with gel electrolyte (Fig 3e in Revision). The comparison showed that the nanofluidic electrode performs better than graphene electrode in gel electrolyte when there is no flooding electrolyte. This experiment does not provide any evidence on the effect of channel alignment on electrode performance. But this comparison does suggest the critical role of the nanofluidic ion channels disregard the alignment. Secondly, we avoid correlating the performance improvement with the channel alignment because we noticed our electrode fabrication method did not let us control the channel alignment. In this case, we cannot provide a nanofluidic electrode without channel alignment to validate the claimed effect.

We believe it is appropriate and justifiable to conclude that the intrinsic nanofluidic ion channels are more dominant to the reported electrode performance. In the Revision, we have corrected the claim and focused more clearly on the effect of nanofluidic structure. However, the structural characterisation still suggests this feature of channel alignment. We prefer to keep it to complete the description of the electrode in the Revision, and we avoid discussing its plausible effect on electrode performance.

Authors' response: Thanks for this comment. We understand the goal of any energy device cannot avoid improving the energy and power density performances. We also stand with the opinion that the device performance is not necessarily the target for a particular paper because there are too many fundamental questions to answer. Some brief discussions on the first and second pages of this Response letter have explained why we do not plan to include the energy/power performance in Revision. More discussions are added below.

In this work, we have measured the full-metric capacitances (gravimetric, volumetric and areal) across a wide range of current density (1 to 30 mA/cm², Figure 3f). We did not use the CV for capacitance measurement. The CV here was merely used to examine if the electrochemical property of the electrodes in liquid and gel electrolytes are consistent.

As to the voltage window, which largely affects the energy density, 0.8 V is indeed lower than those in references 5,6,7,8&10, where the voltages for aqueous and ionic liquid gels are 1 V or above. The voltage in ref 9 is 0.6 V. All references 5-10 are gel-based supercapacitors. In this regard, we understand that the low voltage in our cell configuration will offset the high capacitance of the electrode. However, we also note that it is feasible to widen the 0.8 V to 1 V, or even higher, by adjusting the electrolyte chemistry, as those reported in references 5-10. That means, by applying the reported nanofluidic electrode to high voltage electrolyte, the limits on energy density are resolved. In one ongoing project, we have acquired very high energy density (nearly 100 Wh/L) on 4.5V supercapacitors based on this nanofluidic electrode. We are happy to consider sharing the unpublished results with the reviewer, if required.

As to the calculation of the energy/power density of the devices for either thick or thin electrodes, we did not intend to do so, primarily because it is not the scientific question we would investigate in this work and also because of the low voltage (it is meaningless to consider energy/power density without optimising the voltage).

In short, high voltage capacitor is not the focus of this work because it requires a systematic interfacial modification to enable high voltage compatibility and stability. We would concentrate on the electrode design for solid state gel electrolyte by leveraging the fast nanofluidic ionics in thick electrodes.

Authors' response: We appreciate the comments and agree that supercapacitors are power device. We would also point out the emergent trend of supercapacitors for energy storage. In this work, we propose to use the nanofluidic electrode to enable high capacitance in gel supercapacitors, and to reduce ionic resistance in thick electrodes. Therefore, the concept here holds potential for improving the power and energy performances of supercapacitors for fast-charging energy storage systems.

According to the reviewer's estimation, at 1 mV/s, the charging time is about 13 mins. This time is shorter than normal battery charging. Although this charging time is longer than power supercapacitor for signal filtering, it stores more energy. Therefore, we think there are quite many issues to discuss regarding the pros and cons between power supercapacitor, energy supercapacitor and battery. By any means, it is not a simple 'yes or no' question and it requires substantial efforts to get the right performance configuration in order to meet the practical needs.

Meanwhile, as we clarified above, this work did not use CV and three-electrode cell to measure performance. We used galvanostatic charge discharge and two electrode cell to measure performance, which is widely accepted for supercapacitor material evaluation by both industry and academic researchers. The charging time is determined by the areal current density and the electrode capacitance. In general, short charging relates to low capacitance while high capacitance requires long charging. As shown in Figure 3d-g, the charge time is about 50 s at 10 mA/cm² giving areal capacitance near 5 F/cm² for 50 mg/cm² mass loading. With the 1 mA/cm² charging, the time on 10 mg/cm² mass loading is roughly 750s (Fig. 3d). Indeed, when 1 mA/cm² is applied to 50 mg/cm² electrode, the charging time will be around 1 hour. But the stored charges are almost 5 times to 10 mg/cm² electrode. It is obvious the nanofluidic electrode can operate across a large range of mass loading and areal current density to deliver the nearly proportional capacitances, as shown by the quasi-linear relationships in Figs 3f and 3g. Most importantly, these capacitance behaviours in gel electrolyte do not deviate largely from those in liquid electrolyte (Figs 3d-g and Supplementary Table 3 in Revision), as opposed to the porous graphene electrode. These results demonstrate the potential of the nanofluidic electrodes to meet the variable needs of device configuration: fast charging, high capacitance or balanced conditions. **Still, achieving high capacitance at fast charging is the ultimate challenge for all researchers in this field.** We will continue to work towards this target in our new projects on nanofluidic electrode.

The reviewer commented 'Even conventional activated carbon electrodes can operate at a millimeter thickness at those rates'. Since the reviewer did not list the references for us to compare, we can only explain our understanding on a general base. Activated carbon

electrodes, if the reviewer meant particle or powder samples, have low density ($\sim 0.5 \text{ g/cm}^3$), large portion of micropores (large ionic resistance) and are not common choice for gel supercapacitors. The most relevant example we know is the activated carbon fiber/cloth electrodes, which has to work with high content gel infilling inside the fiber network. Referring to Fig 4b, when high gel filling is present, the overall device gravimetric capacitance will be impaired severely. Therefore, the advantage of nanofluidic electrode is also about increasing the device's capacitance by using less infilling electrolytes. Moreover, the thickness of nanofluidic electrode is less than 0.3mm, only a third of the as-mentioned activated carbon electrodes. In these contexts, the activated carbon electrodes will be less competitive for the gravimetric and volumetric capacitances at the device level.

Authors' response: We appreciate this comment. Indeed, stability is a critical factor to consider. We have extended the cycling test up to 5000 cycles and obtained a retention of 88% (Supplementary Fig 9). The rather stable and high Coulombic Efficiency across the 5000 cycles suggest the reversible process of charging and discharging on TALP nanofluidic electrode. We tried to run for more cycles, but the cells cannot survive and experienced sudden drastic performance drop after about 6000 cycles. We also checked the stability of the TALP material in liquid electrolyte, which was reported in our previous publications (reference 15, 16) and showed capacitance retention around 82% to 85% after 10,000 cycles for a wide range of mass loading. Our previous results suggest the TALP material is chemically stable, as commented by the reviewer. Therefore, we think the plausible reason for the relatively limited cycling life in gel device could be related to the gel itself. We suspect that during the long test, the extra-dry gel could slowly lose water and the loss of water will make the gel a poor ionic conductor at the point of time when the cell cannot operate well.

Authors' response: We appreciate the reviewer's professional standing in laying comparison on appropriate grounds. Our understanding of the 'appropriate comparison' is that one should compare the areal, gravimetric and volumetric capacitances simultaneously, because the practical use of materials is collectively affected by all of the three factors. Claim of high performance of one single metric is unlikely appropriate, despite its popular use in literature. To simultaneously acquire high gravimetric/areal/volumetric capacitances, the trade-off relationships among the electrode density, porosity, mass loading and thickness must be properly resolved. But this is extremely difficult in the system of porous electrodes which always require electrolyte infilling to conduct ions through the pores. For example, to get higher areal capacitance the most popular solution is to make 3D macroporous skeleton to load significant active materials over a large thickness, but this will cause much smaller volumetric capacitance. Or similarly, to get higher gravimetric or volumetric capacitance, one can easily put very little amount active materials (<5 mg/cm²) or make very thin electrode (<1 micron), and apparently, these solutions will lead to negligible areal capacitance. Not reporting one or two capacitance metrics could deliver misleading information and confuse the true performance and feasibility for practical applications. These inappropriate scenarios have been commented by Professor Yury Gogotsi and Professor Patrice Simon in 'True Performance Metrics' for supercapacitor materials. [*Science* 334, 917-918, doi:10.1126/science.1213003 (2011)]. We cited this Perspective in reference 14.

With this work, we want to initiate the comprehensive reliable performance comparison of electrode by reporting areal, gravimetric and volumetric capacitances. This is in particular critical for gel supercapacitors and our nanofluidic electrode design exhibits the potential to solve this performance paradox.

The reviewer mentioned many works that showed high capacitance values. However, because the reviewer gave very limited information of these papers, we are unable to determine whether all of the three capacitance metrics were reported by these works. Therefore, we cannot provide point-by-point responses to those papers mentioned by the reviewers. We are only able to elaborate our perception of the reviewer's intention in general.

In the comparisons in this work, we prefer to choose the publications on gel supercapacitors that have provided all of the three capacitance metrics, or otherwise that have provided sufficient information of electrodes from which we can derive the related capacitances. When we choose the reference work to compare areal capacitance, we set the lower limit of 150 mF/cm², otherwise it will be too low to be meaningful. We acknowledge there are many papers that could have reported higher areal capacitance than our work but those works oftentimes used very thick and porous electrodes and did not report enough information for us to calculate the volumetric and gravimetric capacitances, and some are even not in gel electrolytes. Therefore, these papers were not cited in the performance comparison in Figure 4. When these selection criteria were applied, we were very surprised of the very limited number of comparable references in gel supercapacitors. Nonetheless, we have drawn a comparison among nanofluidic electrodes and porous electrodes in liquid electrolytes (Figure 3h in Revision). In this comparison, we compared the areal capacitance against electrode thickness, to elaborate the saturation of areal capacitance as porous electrode grows thicker. It is not about comparing the capacitance values. – Our revision in main text: By comparison, some porous electrodes usually demonstrate levelling trend as the thickness (mass loading) increases^{5,10,12,31,38,39} (Fig. 3h and Supplementary Fig. 10).

The reviewer commented that coarse-grained porous carbon exhibits high areal performance at thickness close to 0.8 to 1 mm. However, we were not able to find the

relevant paper. We do not know whether this coarse-grained porous carbon was measured in liquid or gel electrolytes, what is the gravimetric or volumetric capacitance of the electrode, what is the areal current density, etc.. We would note that, according to the information given by the reviewer, the thickest nanofluidic electrode in this work is 278 μm , only about 1/3 of the coarse-grained porous carbon. To fairly compare, we expect the coarse-grained porous carbon electrode should possess three time higher areal capacitance in gel electrolyte without gel infiltration (please refer to our analysis in Fig 4b). We will be able to respond more to this comment when the reviewer could provide more details of the related work on coarse-grained porous carbon.

The reviewer commented on vertically aligned MXene or graphene electrode with gel electrolyte demonstrating high areal capacitance. Again, we do not know which papers the reviewer referred to. To our best, we found the most relevant paper that reported the highest volumetric capacitance (over 400 F/cm^3) of MXene electrode with gel electrolyte, which is titled 'Hierarchical Vertically Aligned Titanium Carbide (MXene) Array for Flexible All-Solid-State Supercapacitor with High Volumetric Capacitance' [ACS Appl. Energy Mater. 2019, 2, 9, 6834–6840]. We found that this paper used volumetric current density (0.5 to 8 A/cm^3) and the electrode thickness ranged from 5 to 50 μm . We can work out the areal current density (the unit used in our work) ranged from 2.5 to 40 mA/cm^2 . Taking 10 mA/cm^2 to compare, the volumetric/areal capacitances of our nanofluidic TALP electrode are $\sim 240 \text{ F}/\text{cm}^3$ ($\sim 1.2 \text{ F}/\text{cm}^2$) for electrode thickness of 56 μm (Fig 3 in Revision). This ACS Appl Energy Mater (ACS AEM) paper has these data at 10 mA/cm^2 (2 A/cm^3 for 50 μm electrode, Fig 4c in ACS AEM): $\sim 210 \text{ F}/\text{cm}^3$ ($\sim 1.05 \text{ F}/\text{cm}^2$). These values are not superior to our work. Moreover, this ACS AEM paper did not report the gravimetric performance or provide data that can be used to calculate gravimetric performance. As such, we cannot include it in the comprehensive performance comparison in our revised paper.

The reviewer commented on 'I recall seeing a paper from China reporting MXene-gel electrolyte devices with 10 F/cm^2 capacitance.' We have tried hard to search for the research work from China that reported the areal capacitance of 10 F/cm^2 , but we did not find it. Therefore, we are unable to make unbiased and clear response to this comment.

Again, our innovation is not about a single high capacitance metric, but how to unlock overall capacitance metrics with high values by nanofluidic electrode structure for gel supercapacitors. If this paper from China does report the whole spectrum of capacitance metrics in gel electrolytes, we are surely willing to add it in Fig 4a.

We have read the paper published by Pumera's group on ACS AMI 'MXene-Based Flexible Supercapacitors: Influence of an Organic Ionic Conductor Electrolyte on the Performance' [*ACS Appl. Mater. Interfaces* 2020, 12, 47, 53039–53048]. This paper only provides gravimetric capacitance. It did not provide either volumetric and areal performance or mass loading and density of the electrode. Therefore we cannot calculate the areal and volumetric performance. Thus, we did not put the performance reported in this paper into the performance comparison in our revised paper.

Last, but most importantly, we want to emphasize that this work is not merely about reporting high performance. As clarified in this Response letter and in the Revision, our core scientific breakthrough is the discovery of using nanofluidic electrode to cope with the very common electrode capacitance drop when gel electrolyte is used to replace liquid electrolyte. This performance gap is obviously related to the insufficient ion conduction in gel-filled electrodes. Our study suggests that nanofluidic electrode can be very promising to solve this problem, whereas many previous works implicate it is very challenging with porous electrode design.

Authors' response: First of all, the vertical alignment is accompanied by the horizontal alignment. Both the XRD patterns and the statistical SEM contrast analyses suggest the bi-directional alignment. These data indeed suggest the preference of the vertical alignment but this does not mean the lamellas are all vertically aligned. We have realized some of the description in the original submission were not accurate and have corrected the discussion in the Revision. We also avoided correlating the channel alignment with the performance in the Revision.

As to the reason of alignment, there are two factors to consider. Firstly, the particle is constituted of stacked nanosheets if not delaminated. The pressing used the raw powder without delamination. So the pressure is applied to the particles rather than delaminated nanosheets. Secondly, the particles contain interlayer water between the nanosheets, a

structure similar with clay. Under the pressure, the particles can deform and shear due to the sliding effect of the interlayer water that acts like lubricant. As a result, the pressing condensed the loose particles, which are forced to flow along the uniaxial directions (vertical and horizontal). We do not understand why the vertical alignment was more obvious, but it could be related to the pressing mould which does not provide horizontal pressure and might not produce perfect uniaxial force fields.

We have changed the discussion in Revision as below:

Page 8: **Additional investigation on the lubricating and sliding behavior of TALP particles should be conducted to understand the orientation upon compression.**

Authors' response: Thanks for this comment. We have re-organised the data and put the TEM and SAED to the main text, which will show the flake morphology. We kept the AFM in SI because the information is partly duplicated to TEM. In addition, we have drawn the structure illustration in main text to help understand the material. We have added other data in main text. These are summarized below:

- Reconstruction of 3D structure of TALP particle through reciprocal lattice is added to Figure 1.
- The structure illustration is added to Figure 1.
- SEM image of the cross-section of TALP pellet is added to Figure 2.
- Micro-CT images of TALP pellet is added to Figure 2.
- Comparison of CV curve at 1 mV is added to Figure 3.
- The Nyquist plots of TALP electrode under different temperature is added to Figure 3 and combined with the Arrhenius plot for the nanofluidic ion transport.
- GCD curves of TALP electrode with liquid and gel electrolyte is added to Figure 3d
- For making a clear comparison between the electrode performance when using liquid and gel electrolyte, we added the performance data in liquid electrolyte to the figures, Figure 3 e, f and g.
- **Authors' response:** Thanks for this comment. The interface charge transfer semicircle could be improved by adding highly conductive carbon, carbide or MXenes and we could try these ideas in future projects. One main plausible cause

to the semicircle is the interfacial wettability between the gel and the electrode. Because our electrode is very dense and the gel is very dry, the interfacial ion transport is not as good as those in liquid or wet gels. Therefore, by adopting more conductive additives and most importantly wetting the gel/electrode interface, the charge transfer impedance can be reduced.

- **Authors' response:** Thanks for the comment. We have added the GCD data in Fig 3d.
- We have carefully studied the Mathis paper on Adv Energy Mater (2019, 9, 1902007) and tried to apply the equation to our study. We then noticed a few confusion points (these are discussed briefly as below). To clarify our confusions, we have contacted Dr Tyler S. Mathis at Drexel University, the first author of the AEM paper mentioned by the reviewer. Our discussion with Dr Mathis is ongoing. We are unable to apply this method in our revision before the questions below can be solved.
- A snapshot of the relevant part in Mathis AEM paper is given below:

As was stated in the beginning of Section 3, when the discharge curve of a supercapacitor is linear, the capacitance of the cell can be calculated using the slope of the discharge curve. However, this equation should not be used for a nonideal supercapacitor or for a pseudocapacitive material where the GCD curves are nonlinear. The slope of the curve (dV/dt) for these materials tends to vary during discharging, indicating that the capacitance changes as the voltage changes, even if the kinetics of the electrochemical process are capacitor-like. For example, in Figure 3a the real-time capacitance increases from point 1 to point 3 as the absolute value of the slope of the curve decreases. Therefore, none of the instant capacitance calculations done using the slope from any point on the curve in Figure 3a will accurately represent the charge delivered by cells with this type of discharge curve. The same can be said of the charge portion of this curve. Instead, integration of the areas under the charge and discharge portions of these types of curves should be used for evaluation, following Equation (7)

$$C(\text{F g}^{-1}) = I \int (1/V(t)) dt \quad (7)$$

where I is the applied constant-current density, t is the discharge time, and $V(t)$ is the potential as a function of t .

- We agree with the opinions on the nonlinear GCD profile in the paragraph in Mathis paper. But we have doubts in the equation (7):

- Firstly, according to Mathis paper, $V(t)$ is a function of t and the integration is on dt . In our charge-discharge curve, the voltage window is 0 to 0.8V (Figure 3d). Taking discharge branch, at the end of discharge, the voltage is 0, which derives infinite $1/V(t)|_{t=\text{end of discharge}}$. This condition does not have physical meaning.
- Secondly, if we assume $V(t)$ as the voltage variance (ΔV) for the discharge time of dt , the expression of $I(1/V(t))dt=I(dt)/\Delta V$ means capacitance (C). Equation (7) can be written as $C = \int C$. The integral of capacitance is not defined elsewhere.
- We also found another three relevant papers by S. Zhang, N. Pan, *Adv Energy Mater* (2015, 5, 1401401), M.D. Stoller, R.S. Ruoff, *Energy Environ Sci* (2010, 3, 1294-1301), D.K. Kamporis, X. Ji, E.P. Randviir, C.E. Banks, *RSC Advances* (2015, 5, 12782). We prefer to use the current method ($C=It/V$) because Mathis paper was published very recently and we perceive more publications used the method ($C=It/V$) that is supported by the above latter three papers.
- Nonetheless, we get the point and well understand the reviewer's concern. We have rephased the reported capacitance as the nominal capacitance, which means the average capacitance over the voltage window. We further provided the comparison of Charge (Q) in Supplementary Fig 11 (shown below), which is independent of the linearity of GCD profiles.
- Page 18: Because of the lack of applicable method to calculate the capacitance from non-linear GCD profile, the standard equation for linear GCD profile is used to determine nominal capacitance⁴³⁻⁴⁵. Due to the slight deviation of GCD profile from linearity, these nominal capacitances are considered average over the voltage window. The nominal capacitance of single electrode in this study was calculated from the data acquired with symmetric two-electrode cells.

- Last but not least, we do appreciate the reviewer for pointing out this matter and support Mathis's opinion in his AEM paper. We recognize the importance of calibrating the standard method ($C=It/V$) to derive more accurate capacitance from non-linear GCD profiles. We decided to use this $C=It/V$ method in our work not because it is correct, but it is fairly used by peers for the time being and there is no other better method. How to correctly calculate capacitance from non-linear GCD curve is a problem to solve for the benefit of the community.

Reviewer #2

Authors' response: We appreciate the recognition of the value of this work by the reviewer. We have however revised some of the statements in this revision: 1) we used TALP (tungstate anion linked polyaniline) to replace POAFs. 2) we focused on the nanofluidic structure and tuned down the emphasis on the channel alignment. The reasons have been explained in our responses above.

Authors' response: These are valuable suggestions. We have included more results in the main text to describe the structure of the TALP material (Fig 1). The chemical interaction that holds the TALP structure is hydrogen bond and it was a big challenge to collect high-resolution TEM to reveal the interlayer and intralayer structures. From the current low-mag TEM image below, the interlayer structure is visible when the image is enlarged sufficiently (left). Some resolution will be lost when it is compressed smaller (right).

We put together a descriptive structural model in Figure 1 in the Revision (and below), which is coherent with the conceptual model we described based on chemical analysis in previous publications (ref 15, 16). Briefly, the interlayer structure of TALP consists of dangling tungstate anions that are hydroscopic and have very thin layer of water molecules attached on them. The intralayer structure is constituted of polyaniline chains that are cross-linked by the tungstate anions, the interaction between which is hydrogen bond. The multiple parallel hydrogen bonds between the tungstate and the polyaniline are collectively strong to stabilize the structure.

The stacking of the nanosheets is driven by a collection of hydrogen bonding, electrostatic and ionic interactions. We think the hydrogen bonding could be more dominant because we observed interlayer swelling when the water molecules are replaced with less polar molecules.

The molecular structure of TALP has been discussed to some extent in our previous publications (ref 15,16). The understanding of the nanofluidic feature is the new finding in this work and is critical to the electrode design for gel supercapacitors.

Authors' response: Whether sub-2nm or a particular size of the channels is the best is dependent on the specific electrolyte chemistry, including the ion size, solvation and solvent molecule size. When we discussed sub-2nm in the original version, it is suggested by many recent publications (ref 10, 12, 19, 20, 21) based on the collective experimental observation of a range of supercapacitor materials. The real science governing this phenomenon is not fully understood but it is probably relevant to the overlapping of the electric double layer (EDL) on the opposite surfaces of the channel. The overlapped EDL could produce unified directional electric field to speed up the ion transport. We understand the uncertainty in regards of the optimal size for ion transport and we realize the channel size is not dominant in the experimental findings reported in this work. As such, we refined our description and elaboration of the results to the nanofluidic structure. The terminology of sub-2nm channel is now merely used as descriptive data to understand the dimension/geometry of the nanofluidic channel and we avoided correlating the channel size with the performance.

With high pressure compression, or other mechanical approaches, the channel size can be tuned. We have obtained some interesting data on how the wider channels can make the ion conduction faster. Those results are submitted for review with another journal. We could consider sharing the draft if required.

As to ‘the comparison with other samples to prove the ion transport’, we compared the TALP with porous graphene in Figure 3e the Revision (and below). The results are very interesting and are discussed above. Here is a summary. In liquid electrolyte cell, where the electrodes are flooded by electrolyte, both TALP and graphene electrodes perform reasonably well. In gel electrolyte, where the electrodes are not flooded, the TALP performs similarly with its behaviour in liquid electrolyte whereas graphene’s performance drops drastically. We treat this contrast electrode behaviour as the clear sign to prove the function of nanofluidic ion channels for gel supercapacitor.

Authors’ response: Thanks for the suggestion. We updated the calculation methods (Supplementary Fig 6) and the equations in main text (Page 9 and Page 11 in Revision).

Reviewer #3

Authors’ response: Thanks for considering this work. As above statement, POAFs is replaced with TALP to make the description more accurate and meaningful.

Authors’ response: Thanks for this comment. As discussed in the Revision, this work is not about the synthesis of the TALP material. We reported the synthesis of TALP in ref 15 and 16.

This work has the following breakthroughs: 1) combine the TEM and NMR analyses to discover the nanofluidic structure of TALP; 2) propose the use of nanofluidic structure to enhance the electrode capacitance in gel electrolyte to the level in liquid electrolyte; 3) demonstrate the impact of applying nanofluidic electrode on the gel supercapacitor performance.

Despite the use of the same materials as our previous publications, this manuscript reports the new, more profound understanding and more impactful application of the TALP materials for quasi-solid-state or solid state energy storage devices.

We understand the innovation in materials synthesis in this work is minimal. However, the innovation of new properties and new application potentials are magnificent in this work.

Authors' response: Thanks for this comment. As we discussed above, the alignment is bi-directional and is more inclined to the vertical direction. This process is not controllable in our current pressing setup. We think extra mechanical experiments are necessary to fully understand the lubricating and sliding behaviour of the TALP particles, which should well deserve an independent study. To prevent misunderstanding, we avoided correlating the alignment with the electrode performance. We might revisit this plausible effect if we can control the alignment of the channels.

Authors' response: Thanks for this comment. In our original submission, we only compared the capacitance of gel-based electrode. As we discussed above, there are very limited number of publications that report the areal, gravimetric and volumetric capacitances and the ref 10 (a Nature Energy paper) is one of these. Moreover, the gravimetric and volumetric capacitances in this ref 10 are superior to many other even in non-IL gel electrolytes (Fig 4a). When we compare, we do not target on a single paper, but on a group of porous electrodes. This ref 10 is one typical example of graphene/graphene oxide laminar composite with tuneable pore size. On account of these thoughts, omitting this ref 10 seems inappropriate either.

To avoid misunderstanding, we have added a short explaining sentence in the figure captions:

Fig 3&4: *Note that ionogel and ionic liquid electrolytes used in reference 10.*

We also compared the Fig 4a based on Charge (see above response to Reviewer 1 and Supplementary Fig 11).

As to the pseudocapacitive examples, we didn't find sufficient data in related works that allow us to calculate the three capacitance metrics in gel supercapacitors. Therefore, we are unable to put those in the comparison. The compared example includes only ref. 9 (MXene) in Fig 4a.

Lastly, performance comparison in our work is not purposed to reporting record high value. We focus on how the nanofluidic electrode design could introduce value-adding advances to the gel supercapacitors. The difference in capacitance is the experimental evidence that support our theory. Our findings could be interpreted in more meaningful ways: 1) nanofluidic TALP electrode can balance areal, gravimetric and volumetric capacitances without obvious constraint on one single metric; 2) nanofluidic TALP electrode can reduce the use of infilling gel and hence reduce the total volume and mass of the device; 3) nanofluidic TALP can maintain comparable capacitance in gel to that in liquid.

Authors' response: We have added the specific surface area data ($\sim 22 \text{ m}^2/\text{g}$). Due to the very low porosity, the N_2 cryo-sorption analysis does not derive the pore size distribution. We have used microCT to measure the total porosity of the compacted TALP pellet (Fig 2).

Authors' response: These data are added in Supplementary Table 3.

	Mass loading (mg cm^{-2})	Thickness (μm)	Areal capacitance (F cm^{-2})		
			1 m A cm^{-2}	10 m A cm^{-2}	30 m A cm^{-2}
Liquid electrolyte	10	56	1.80	1.28	0.89
	20	111	3.45	2.35	1.61
	30	167	5.25	3.57	2.39
	40	222	6.99	4.50	2.89
	50	278	9.10	6.28	3.92
Gel electrolyte	10	56	1.84	1.27	0.72
	20	111	3.59	2.36	1.40
	30	167	5.38	3.29	1.88
	40	222	7.23	4.25	2.37
	50	278	8.94	5.30	3.06

Reviewers'

Comments:

Reviewer

#1:

Remarks

to

the

Author:

The authors performed a thorough revision of the manuscript. While I still have concerns regarding the low charge-discharge rate and application of a linear equation to nonlinear galvanostatic discharge curves, I feel that the manuscript has enough novelty and contains sufficient basic science to justify publication.

Reviewer

#2:

Remarks

to

the

Author:

In the revised manuscript, the authors have made additional efforts to address the issues raised by the reviewers. Most of my concerns have been resolved, except for the quality of the TEM image demonstrating the interlayer/intralayer nanofluid structures of TALP. The TEM image presented is not of high enough quality in my opinion. Nevertheless, I leave it to the editor to decide if additional effort is needed.

Minor

comments:

-It would be helpful if the authors could include the values of the achieved areal/volumetric/gravimetric

capacity in the abstract.

-Since the authors can control the size of the nanofluid channels and achieve even better capacitance performance, it would be good if the authors include this perspective/statement in the conclusion section.

Reviewer #3:

Remarks to the Author:

Notable changes have been made in this revised manuscript. Considering that the electrode materials have been reported in their previous reports, here the authors modified their focus on the nanofluidic structure of electrode for enhanced capacitance in gel electrolytes. It is the case that the TALP electrode in Na₂SO₄-based gel electrolyte exhibits a good balance of areal, gravimetric and volumetric capacitances even at very high mass loadings. My major reserved comment is the inappropriate comparison of capacitances of TALP electrode with other porous electrodes. Since the TALP is a hybrid of tungstate and polyaniline with a low specific surface area ($\sim 22 \text{ m}^2/\text{g}$), the capacitance of TALP electrode should be dominated by pseudocapacitance but not EDL capacitance. However, except for reference 9, the compared references (5-8, 10, 12 31, 38, and 39 in Fig. 3e, 3h, and 4) utilized EDLC-type materials (graphene or activated carbon). It is misleading to compare the capacitance of two-type electrode materials of different storage mechanisms, especially under the condition that the power and cycling performances of TALP electrode are inferior to the EDLC-type materials. Moreover, as mentioned in my previous version, comparison of electrode materials in aqueous and non-aqueous electrolytes also lacks rigor. The authors attribute the inappropriate comparison to "the very limited number of comparable references in gel supercapacitors". In my opinion, it is not necessary to "choose the publications on gel supercapacitors that have provided all of the three capacitance metrics". Otherwise, it is better to delete the related comparative figures rather than displaying them in the main text. As minor comment, I agree with Review 1's comments to show energy/power densities, and to calculate the capacitance by integration.

Responses to REVIEWER COMMENTS

Summary

While leaving our detailed point-by-point responses in a later part, we would like to summarize a few major changes here.

1) The description of the core finding in this work is refined.

We rediscovered the scientific value of the work and refocused it on the use of nanofluidic electrode to enhance electrode capacitance in gel electrolytes.

The primary scientific challenge of solid-state gel supercapacitors lies in the limiting ion accessible electrode/electrolyte interface in gel electrolyte. In our work, the most valuable finding is the use of nanofluidic channels to facilitate the ion transport in bulk dense electrode through the interlayer nanoconfined water, which allows the acquisition of the very close capacitance in gel electrolyte to that in liquid electrolyte. The minimal gel-to-liquid capacitance gap for nanofluidic electrode is oftentimes unachievable for conventional porous electrodes, and typical porous graphene electrode was taken as the control to validate this result in the revision (Fig. 3e).

This unique nanofluidic electrode also eliminate the mandatory requirement of infilling porous electrode with gel or sol electrolytes. Benefiting from the superdense and ion percolating interior, the nanofluidic electrode delivers coherently high capacitance values on the three metrics of areal, gravimetric and volumetric.

2) The title is revised to keep coherent with the refined research focus.

Old title: Superdense solid state energy storage by aligned nanofluidic frameworks

New title: Capacitance enhancement by nanofluidic voidless electrodes in solid state gel supercapacitors

The *New title* eliminates two claims in the *Old title*, including the aligned nanofluidic frameworks and solid-state energy storage. We believe the New title will avoid misleading information and accurately reflect the core scientific breakthrough in this work (detailed analysis of the scientific finding is given as above).

- Reason not to claim 'nanofluidic alignment' in title

We performed extra experiments to validate the two highly relevant factors of the nanofluidic electrode: presence of nanofluidic channels, and channel

alignment. For the first factor, we adopted a non-nanofluidic porous graphene electrode as a control to understand the role of nanofluidic channels on the electrode capacitance in gel electrolyte. As data shown in Revision (Fig. 3e), we are confident to conclude the nanofluidic channels are dominant in minimizing the capacitance difference between gel and liquid electrolyte. We also tried different pressing methods but unluckily we were unable to control the nature of the alignment precisely. As such, we cannot acquire a nanofluidic electrode with different alignment or without alignment. Therefore, we cannot validate if the aligned nanofluidic channels are dominant in the study. Note that we still describe the channel alignment in the Revision (Fig. 2) as a unique feature of the electrode, but we do not claim its dominant correlation with the performance.

- Reason not to highlight 'superdense energy storage' in title

Claim of 'superdense energy storage' could lead perception of high energy density and we decided to avoid using this term in Revision. In the Revision, we only consider the capacitance enhancement because it is directly relevant to the electrode/electrolyte interface accessibility to ions, which is widely regarded the most fundamental electrode mechanism of supercapacitor. Our study devotes to solving the ion accessibility in gel solid-state supercapacitors by nanofluidic electrode and our results affirm the feasibility and advancements of this new idea.

We anticipate designing high-performance gel supercapacitors based on this nanofluidic electrode concept in our next project through optimising a series of device parameters (electrolyte, voltage, impedance, etc.) that influence both energy and power performances. We feel that the focus of superior device (energy/power) performances in this work will probably distract the attention from the nanofluidic electrode and hence dilute the scientific impact. We believe highlighting the conceptual breakthrough in the nanofluidic electrode design in gel supercapacitors is way more impactful and rational based on our current results.

3) We abandoned the terminology of POAF (polymer-oxyanion framework)

The so-called POAF in the original submission is too general and cannot accurately relate to the actual material used in this study. In Revision, we decided

to use TALP that is reported in our previous work (ref 15, 16). We hope this change will let the material easy to understand.

Reviewer #1 (Remarks to the Author):

In this article, the authors demonstrate compact capacitive energy storage in a solid-state electrochemical capacitor with thick electrodes. The authors have achieved bidirectional dense alignment of sub-2 nm nanofluidic ion channels in compacted electrodes. The authors assign it to nanofluidic superionic conduction that is 2-4 orders of magnitude faster than in solid gel electrolytes. They used two 2D conductive and hydrophilic polymer-oxyanion frameworks (POAFs) to construct highly aligned ion channels because of their clay-like structure and rheology. This is an interesting idea, but I'm not convinced if the system under study lives to expectations. I cannot recommend the paper for publication, at least not in the current state.

Authors' response: These comments are valuable. We have carefully reinvestigated the data and conducted more control experiments. Based on the new findings, we have made major changes in the Revision and these are summarized here.

We revisited the claim of 'highly aligned ion channels'. We now consider it a less significant and unproven factor for the electrode behaviour in solid state gel electrolyte. Our rationales are based on two reasons. Firstly, we conducted a comparative test on porous graphene electrode (made from thermal reduction of reduced graphene oxide) with gel electrolyte (Fig 3e in Revision). The comparison showed that the nanofluidic electrode performs better than graphene electrode in gel electrolyte when there is no flooding electrolyte. This experiment does not provide any evidence on the effect of channel alignment on electrode performance. But this comparison does suggest the critical role of the nanofluidic ion channels disregard the alignment. Secondly, we avoid correlating the performance improvement with the channel alignment because we noticed our electrode fabrication method did not let

us control the channel alignment. In this case, we cannot provide a nanofluidic electrode without channel alignment to validate the claimed effect.

We believe it is appropriate and justifiable to conclude that the intrinsic nanofluidic ion channels are more dominant to the reported electrode performance. In the Revision, we have corrected the claim and focused more clearly on the effect of nanofluidic structure. However, the structural characterisation still suggests this feature of channel alignment. We prefer to keep it to complete the description of the electrode in the Revision, and we avoid discussing its plausible effect on electrode performance.

While the obtained capacitance values are high, it's important to remember that the voltage window is narrow (0.8 V), the charge/discharge rate is very low (1 mV/s in CV) and the energy and power density values are going to be low as well. Have the authors calculated energy and power density for their devices, even those with relatively thin electrodes?

Authors' response: Thanks for this comment. We understand the goal of any energy device cannot avoid improving the energy and power density performances. We also stand with the opinion that the device performance is not necessarily the target for a particular paper because there are too many fundamental questions to answer. Some brief discussions on the first and second pages of this Response letter have explained why we do not plan to include the energy/power performance in Revision. More discussions are added below.

In this work, we have measured the full-metric capacitances (gravimetric, volumetric and areal) across a wide range of current density (1 to 30 mA/cm², Figure 3f). We did not use the CV for capacitance measurement. The CV here was merely used to examine if the electrochemical property of the electrodes in liquid and gel electrolytes are consistent.

As to the voltage window, which largely affects the energy density, 0.8 V is indeed lower than those in references 5,6,7,8&10, where the voltages for aqueous and ionic liquid gels are 1 V or above. The voltage in ref 9 is 0.6 V. All references 5-10 are gel-based supercapacitors. In this regard, we understand that the low voltage in our cell configuration will offset the high capacitance of the electrode. However, we also note that it is feasible to widen the 0.8 V to 1 V, or even higher, by adjusting the

electrolyte chemistry, as those reported in references 5-10. That means, by applying the reported nanofluidic electrode to high voltage electrolyte, the limits on energy density are resolved. In one ongoing project, we have acquired very high energy density (nearly 100 Wh/L) on 4.5V supercapacitors based on this nanofluidic electrode. We are happy to consider sharing the unpublished results with the reviewer, if required.

As to the calculation of the energy/power density of the devices for either thick or thin electrodes, we did not intend to do so, primarily because it is not the scientific question we would investigate in this work and also because of the low voltage (it is meaningless to consider energy/power density without optimising the voltage).

In short, high voltage capacitor is not the focus of this work because it requires a systematic interfacial modification to enable high voltage compatibility and stability. We would concentrate on the electrode design for solid state gel electrolyte by leveraging the fast nanofluidic ionics in thick electrodes.

It's important to remember that supercapacitors are power devices. If the power is low, they have no advantage over batteries. However, CVs were recorded at 1 mV/s (800 min to charge = >13 min). GCD curves show charging in excess of an hour. These charging rates are too low, even for gel electrolyte supercaps. Even conventional activated carbon electrodes can operate at a millimeter thickness at those rates. How can this issue be addressed?

Authors' response: We appreciate the comments and agree that supercapacitors are power device. We would also point out the emergent trend of supercapacitors for energy storage. In this work, we propose to use the nanofluidic electrode to enable high capacitance in gel supercapacitors, and to reduce ionic resistance in thick electrodes. Therefore, the concept here holds potential for improving the power and energy performances of supercapacitors for fast-charging energy storage systems.

According to the reviewer's estimation, at 1 mV/s, the charging time is about 13 mins. This time is shorter than normal battery charging. Although this charging time is longer than power supercapacitor for signal filtering, it stores more energy. Therefore, we think there are quite many issues to discuss regarding the pros and cons between power supercapacitor, energy supercapacitor and battery. By any

means, it is not a simple 'yes or no' question and it requires substantial efforts to get the right performance configuration in order to meet the practical needs.

Meanwhile, as we clarified above, this work did not use CV and three-electrode cell to measure performance. We used galvanostatic charge discharge and two electrode cell to measure performance, which is widely accepted for supercapacitor material evaluation by both industry and academic researchers. The charging time is determined by the areal current density and the electrode capacitance. In general, short charging relates to low capacitance while high capacitance requires long charging. As shown in Figure 3d-g, the charge time is about 50 s at 10 mA/cm² giving areal capacitance near 5 F/cm² for 50 mg/cm² mass loading. With the 1 mA/cm² charging, the time on 10 mg/cm² mass loading is roughly 750s (Fig. 3d). Indeed, when 1 mA/cm² is applied to 50 mg/cm² electrode, the charging time will be around 1 hour. But the stored charges are almost 5 times to 10 mg/cm² electrode. It is obvious the nanofluidic electrode can operate across a large range of mass loading and areal current density to deliver the nearly proportional capacitances, as shown by the quasi-linear relationships in Figs 3f and 3g. Most importantly, these capacitance behaviours in gel electrolyte do not deviate largely from those in liquid electrolyte (Figs 3d-g and Supplementary Table 3 in Revision), as opposed to the porous graphene electrode. These results demonstrate the potential of the nanofluidic electrodes to meet the variable needs of device configuration: fast charging, high capacitance or balanced conditions. **Still, achieving high capacitance at fast charging is the ultimate challenge for all researchers in this field.** We will continue to work towards this target in our new projects on nanofluidic electrode.

The reviewer commented 'Even conventional activated carbon electrodes can operate at a millimeter thickness at those rates'. Since the reviewer did not list the references for us to compare, we can only explain our understanding on a general base. Activated carbon electrodes, if the reviewer meant particle or powder samples, have low density (~0.5 g/cm³), large portion of micropores (large ionic resistance) and are not common choice for gel supercapacitors. The most relevant example we know is the activated carbon fiber/cloth electrodes, which has to work with high content gel infilling inside the fiber network. Referring to Fig 4b, when high gel filling is present, the overall device gravimetric capacitance will be impaired severely.

Therefore, the advantage of nanofluidic electrode is also about increasing the device's capacitance by using less infilling electrolytes. Moreover, the thickness of nanofluidic electrode is less than 0.3mm, only a third of the as-mentioned activated carbon electrodes. In these contexts, the activated carbon electrodes will be less competitive for the gravimetric and volumetric capacitances at the device level.

A 10% loss of performance after 4000 cycles is not very impressive either. It's actually quite typical for pseudocapacitive electrode to show about 10,000 cycles lifetime (till 80% of the initial capacitance value). What is the main mechanism of degradation?

Authors' response: We appreciate this comment. Indeed, stability is a critical factor to consider. We have extended the cycling test up to 5000 cycles and obtained a retention of 88% (Supplementary Fig 9). The rather stable and high Coulombic Efficiency across the 5000 cycles suggest the reversible process of charging and discharging on TALP nanofluidic electrode. We tried to run for more cycles, but the cells cannot survive and experienced sudden drastic performance drop after about 6000 cycles. We also checked the stability of the TALP material in liquid electrolyte, which was reported in our previous publications (reference 15, 16) and showed capacitance retention around 82% to 85% after 10,000 cycles for a wide range of mass loading. Our previous results suggest the TALP material is chemically stable, as commented by the reviewer. Therefore, we think the plausible reason for the relatively limited cycling life in gel device could be related to the gel itself. We suspect that during the long test, the extra-dry gel could slowly lose water and the loss of water will make the gel a poor ionic conductor at the point of time when the cell cannot operate well.

The authors claim that the areal and volumetric capacitance can be, respectively, 50 and 15 times higher compared to conventional porous supercap electrodes. Coarse-grained porous carbon electrodes with thickness up to 0.8-1 mm have been reported and showed comparable areal capacitance at a significantly higher rate in organic electrolytes. Also, graphene and MXene were vertically aligned to produce thick electrodes operating at significantly higher rates in liquid and gel electrolytes. I recall seeing a paper from China reporting MXene-gel electrolyte devices with 10 F/cm² capacitance. Pumera's group published high values of capacitance for MXene-ionogel devices in ACS AMI recently. An appropriate comparison should be provided.

Authors' response: We appreciate the reviewer's professional standing in laying comparison on appropriate grounds. Our understanding of the 'appropriate comparison' is that one should compare the areal, gravimetric and volumetric capacitances simultaneously, because the practical use of materials is collectively affected by all of the three factors. Claim of high performance of one single metric is unlikely appropriate, despite its popular use in literature. To simultaneously acquire high gravimetric/areal/volumetric capacitances, the trade-off relationships among the electrode density, porosity, mass loading and thickness must be properly resolved. But this is extremely difficult in the system of porous electrodes which always require electrolyte infilling to conduct ions through the pores. For example, to get higher areal capacitance the most popular solution is to make 3D macroporous skeleton to load significant active materials over a large thickness, but this will cause much smaller volumetric capacitance. Or similarly, to get higher gravimetric or volumetric capacitance, one can easily put very little amount active materials (<5 mg/cm²) or make very thin electrode (<1 micron), and apparently, these solutions will lead to negligible areal capacitance. Not reporting one or two capacitance metrics could deliver misleading information and confuse the true performance and feasibility for practical applications. These inappropriate scenarios have been commented by Professor Yury Gogotsi and Professor Patrice Simon in 'True Performance Metrics' for supercapacitor materials. [*Science* 334, 917-918, doi:10.1126/science.1213003 (2011)]. We cited this Perspective in reference 14.

With this work, we want to initiate the comprehensive reliable performance comparison of electrode by reporting areal, gravimetric and volumetric capacitances.

This is in particular critical for gel supercapacitors and our nanofluidic electrode design exhibits the potential to solve this performance paradox.

The reviewer mentioned many works that showed high capacitance values. However, because the reviewer gave very limited information of these papers, we are unable to determine whether all of the three capacitance metrics were reported by these works. Therefore, we cannot provide point-by-point responses to those papers mentioned by the reviewers. We are only able to elaborate our perception of the reviewer's intention in general.

In the comparisons in this work, we prefer to choose the publications on gel supercapacitors that have provided all of the three capacitance metrics, or otherwise that have provided sufficient information of electrodes from which we can derive the related capacitances. When we choose the reference work to compare areal capacitance, we set the lower limit of 150 mF/cm², otherwise it will be too low to be meaningful. We acknowledge there are many papers that could have reported higher areal capacitance than our work but those works oftentimes used very thick and porous electrodes and did not report enough information for us to calculate the volumetric and gravimetric capacitances, and some are even not in gel electrolytes. Therefore, these papers were not cited in the performance comparison in Figure 4. When these selection criteria were applied, we were very surprised of the very limited number of comparable references in gel supercapacitors. Nonetheless, we have drawn a comparison among nanofluidic electrodes and porous electrodes in liquid electrolytes (Figure 3h in Revision). In this comparison, we compared the areal capacitance against electrode thickness, to elaborate the saturation of areal capacitance as porous electrode grows thicker. It is not about comparing the capacitance values. – Our revision in main text: By comparison, some porous electrodes usually demonstrate levelling trend as the thickness (mass loading) increases^{5,10,12,31,38,39} (Fig. 3h and Supplementary Fig. 10).

The reviewer commented that coarse-grained porous carbon exhibits high areal performance at thickness close to 0.8 to 1 mm. However, we were not able to find the relevant paper. We do not know whether this coarse-grained porous carbon was measured in liquid or gel electrolytes, what is the gravimetric or volumetric capacitance of the electrode, what is the areal current density, etc.. We would note that, according to the information given by the reviewer, the thickest nanofluidic

electrode in this work is 278 μm , only about 1/3 of the coarse-grained porous carbon. To fairly compare, we expect the coarse-grained porous carbon electrode should possess three time higher areal capacitance in gel electrolyte without gel infiltration (please refer to our analysis in Fig 4b). We will be able to respond more to this comment when the reviewer could provide more details of the related work on coarse-grained porous carbon.

The reviewer commented on vertically aligned MXene or graphene electrode with gel electrolyte demonstrating high areal capacitance. Again, we do not know which papers the reviewer referred to. To our best, we found the most relevant paper that reported the highest volumetric capacitance (over 400 F/cm^3) of MXene electrode with gel electrolyte, which is titled 'Hierarchical Vertically Aligned Titanium Carbide (MXene) Array for Flexible All-Solid-State Supercapacitor with High Volumetric Capacitance' [ACS Appl. Energy Mater. 2019, 2, 9, 6834–6840]. We found that this paper used volumetric current density (0.5 to 8 A/cm^3) and the electrode thickness ranged from 5 to 50 μm . We can work out the areal current density (the unit used in our work) ranged from 2.5 to 40 mA/cm^2 . Taking 10 mA/cm^2 to compare, the volumetric/areal capacitances of our nanofluidic TALP electrode are $\sim 240 \text{ F}/\text{cm}^3$ ($\sim 1.2 \text{ F}/\text{cm}^2$) for electrode thickness of 56 μm (Fig 3 in Revision). This ACS Appl Energy Mater (ACS AEM) paper has these data at 10 mA/cm^2 (2 A/cm^3 for 50 μm electrode, Fig 4c in ACS AEM): $\sim 210 \text{ F}/\text{cm}^3$ ($\sim 1.05 \text{ F}/\text{cm}^2$). These values are not superior to our work. Moreover, this ACS AEM paper did not report the gravimetric performance or provide data that can be used to calculate gravimetric performance. As such, we cannot include it in the comprehensive performance comparison in our revised paper.

The reviewer commented on 'I recall seeing a paper from China reporting MXene-gel electrolyte devices with 10 F/cm^2 capacitance.' We have tried hard to search for the research work from China that reported the areal capacitance of 10 F/cm^2 , but we did not find it. Therefore, we are unable to make unbiased and clear response to this comment. Again, our innovation is not about a single high capacitance metric, but how to unlock overall capacitance metrics with high values by nanofluidic electrode structure for gel supercapacitors. If this paper from China does report the whole spectrum of capacitance metrics in gel electrolytes, we are surely willing to add it in Fig 4a.

We have read the paper published by Pumera's group on ACS AMI 'MXene-Based Flexible Supercapacitors: Influence of an Organic Ionic Conductor Electrolyte on the Performance' [*ACS Appl. Mater. Interfaces* 2020, 12, 47, 53039–53048]. This paper only provides gravimetric capacitance. It did not provide either volumetric and areal performance or mass loading and density of the electrode. Therefore we cannot calculate the areal and volumetric performance. Thus, we did not put the performance reported in this paper into the performance comparison in our revised paper.

Last, but most importantly, we want to emphasize that this work is not merely about reporting high performance. As clarified in this Response letter and in the Revision, our core scientific breakthrough is the discovery of using nanofluidic electrode to cope with the very common electrode capacitance drop when gel electrolyte is used to replace liquid electrolyte. This performance gap is obviously related to the insufficient ion conduction in gel-infilled electrodes. Our study suggests that nanofluidic electrode can be very promising to solve this problem, whereas many previous works implicate it is very challenging with porous electrode design.

Why is the vertical alignment of lamellas was maintained after uniaxial pressing (schematic in Fig. 2a)? 2D flakes usually align in-plane, normal to the applied force, upon compression.

Authors' response: First of all, the vertical alignment is accompanied by the horizontal alignment. Both the XRD patterns and the statistical SEM contrast analyses suggest the bi-directional alignment. These data indeed suggest the preference of the vertical alignment but this does not mean the lamellas are all vertically aligned. We have realized some of the description in the original submission were not accurate and have corrected the discussion in the Revision. We also avoided correlating the channel alignment with the performance in the Revision.

As to the reason of alignment, there are two factors to consider. Firstly, the particle is constituted of stacked nanosheets if not delaminated. The pressing used the raw powder without delamination. So the pressure is applied to the particles rather than delaminated nanosheets. Secondly, the particles contain interlayer water between the nanosheets, a structure similar with clay. Under the pressure, the particles can deform and shear due to the sliding effect of the interlayer water that acts like

lubricant. As a result, the pressing condensed the loose particles, which are forced to flow along the uniaxial directions (vertical and horizontal). We do not understand why the vertical alignment was more obvious, but it could be related to the pressing mould which does not provide horizontal pressure and might not produce perfect uniaxial force fields.

We have changed the discussion in Revision as below:

Page 8: **Additional investigation on the lubricating and sliding behavior of TALP particles should be conducted to understand the orientation upon compression.**

Some of the material characterization data, such as AFM analysis of flakes and photos of pellets should be moved to main text. This will help readers to understand the material under study. It's difficult to read and understand the article without having SI open simultaneously.

Authors' response: Thanks for this comment. We have re-organised the data and put the TEM and SAED to the main text, which will show the flake morphology. We kept the AFM in SI because the information is partly duplicated to TEM. In addition, we have drawn the structure illustration in main text to help understand the material. We have added other data in main text. These are summarized below:

- Reconstruction of 3D structure of TALP particle through reciprocal lattice is added to Figure1.
- The structure illustration is added to Figure 1.
- SEM image of the cross-section of TALP pellet is added to Figure 2.
- Micro-CT images of TALP pellet is added to Figure 2.
- Comparison of CV curve at 1 mV is added to Figure 3.
- The Nyquist plots of TALP electrode under different temperature is added to Figure 3 and combined with the Arrhenius plot for the nanofluidic ion transport.
- GCD curves of TALP electrode with liquid and gel electrolyte is added to Figure 3d
- For making a clear comparison between the electrode performance when using liquid and gel electrolyte, we added the performance data in liquid electrolyte to the figures, Figure 3 e, f and g.

The Nyquist plots suggest that the ionic conductivity is good, but there are large semicircles. Can the charge transfer problem be resolved by adding materials with high electronic conductivity, such as 2D Ti₃C₂ or rGO?

Authors' response: Thanks for this comment. The interface charge transfer semicircle could be improved by adding highly conductive carbon, carbide or MXenes and we could try these ideas in future projects. One main plausible cause to the semicircle is the interfacial wettability between the gel and the electrode. Because our electrode is very dense and the gel is very dry, the interfacial ion transport is not as good as those in liquid or wet gels. Therefore, by adopting more conductive additives and most importantly wetting the gel/electrode interface, the charge transfer impedance can be reduced.

The authors report that GCD data (Fig. S9) were used for capacitance calculations. They should be reported in the main manuscript, along with CVs and Nyquist plots. Due to nonlinear discharge curves and non-rectangular CVs, the authors need to integrate (see, e.g, T.S Mathis in Adv. En. Mater., 2019). The equation for areal capacitance shown on page 18 was derived for an ideal capacitor. Therefore, I recommend recalculating all capacitance values.

Authors' response: Thanks for the comment. We have added the GCD data in Fig 3d.

We have carefully studied the Mathis paper on Adv Energy Mater (2019, 9, 1902007) and tried to apply the equation to our study. We then noticed a few confusion points (these are discussed briefly as below). To clarify our confusions, we have contacted Dr Tyler S. Mathis at Drexel University, the first author of the AEM paper mentioned by the reviewer. Our discussion with Dr Mathis is ongoing. We are unable to apply this method in our revision before the questions below can be solved.

A snapshot of the relevant part in Mathis AEM paper is given below:

As was stated in the beginning of Section 3, when the discharge curve of a supercapacitor is linear, the capacitance of the cell can be calculated using the slope of the discharge curve. However, this equation should not be used for a nonideal supercapacitor or for a pseudocapacitive material where the GCD curves are nonlinear. The slope of the curve (dV/dt) for these materials tends to vary during discharging, indicating that the capacitance changes as the voltage changes, even if the kinetics of the electrochemical process are capacitor-like. For example, in Figure 3a the real-time capacitance increases from point 1 to point 3 as the absolute value of the slope of the curve decreases. Therefore, none of the instant capacitance calculations done using the slope from any point on the curve in Figure 3a will accurately represent the charge delivered by cells with this type of discharge curve. The same can be said of the charge portion of this curve. Instead, integration of the areas under the charge and discharge portions of these types of curves should be used for evaluation, following Equation (7)

$$C(\text{F g}^{-1}) = I \int (1/V(t)) dt \quad (7)$$

where I is the applied constant-current density, t is the discharge time, and $V(t)$ is the potential as a function of t .

We agree with the opinions on the nonlinear GCD profile in the paragraph in Mathis paper. But we have doubts in the equation (7):

Firstly, according to Mathis paper, $V(t)$ is a function of t and the integration is on dt . In our charge-discharge curve, the voltage window is 0 to 0.8V (Figure 3d). Taking discharge branch, at the end of discharge, the voltage is 0, which derives infinite $1/V(t)|_{t=\text{end of discharge}}$. This condition does not have physical meaning.

Secondly, if we assume $V(t)$ as the voltage variance (ΔV) for the discharge time of dt , the expression of $I(1/V(t))dt=I(dt)/\Delta V$ means capacitance (C). Equation (7) can be written as $C = \int C$. The integral of capacitance is not defined elsewhere.

We also found another three relevant papers by S. Zhang, N. Pan, Adv Energy Mater (2015, 5, 1401401), M.D. Stoller, R.S. Ruoff, Energy Environ Sci (2010, 3, 1294-1301), D.K. Kamporis, X. Ji, E.P. Randviir, C.E. Banks, RSC Advances (2015, 5, 12782). We prefer to use the current method ($C=It/V$) because Mathis paper was published very recently and we perceive more publications used the method ($C=It/V$) that is supported by the above latter three papers.

Nonetheless, we get the point and well understand the reviewer's concern. We have rephased the reported capacitance as the nominal capacitance, which means the average capacitance over the voltage window. We further provided the comparison

of Charge (Q) in Supplementary Fig 11 (shown below), which is independent of the linearity of GCD profiles.

Page 18: Because of the lack of applicable method to calculate the capacitance from non-linear GCD profile, the standard equation for linear GCD profile is used to determine nominal capacitance⁴³⁻⁴⁵. Due to the slight deviation of GCD profile from linearity, these nominal capacitances are considered average over the voltage window. The nominal capacitance of single electrode in this study was calculated from the data acquired with symmetric two-electrode cells.

Last but not least, we do appreciate the reviewer for pointing out this matter and support Mathis's opinion in his AEM paper. We recognize the importance of calibrating the standard method ($C=It/V$) to derive more accurate capacitance from non-linear GCD profiles. We decided to use this $C=It/V$ method in our work not because it is correct, but it is fairly used by peers for the time being and there is no other better method. How to correctly calculate capacitance from non-linear GCD curve is a problem to solve for the benefit of the community.

Reviewer #2 (Remarks to the Author):

The development of high volumetric and high areal energy densities has been an attractive objective for the current pursuit of solid-state supercapacitors, which can be promising for powering different miniaturized electronic components. However, the current strategies utilized for porous electrode materials have failed to provide high energy densities when solid-state electrolytes are employed, which are mostly due to the limited ion diffusion in solid-state devices. In this work, the authors present the superdense solid-state energy storage using aligned nanofluidic frameworks

based on POAFs. They have achieved excellent solid-state capacitance per footprint area and volume, which are superior to the state-of-art porous electrode materials. Overall, the achieved results are interesting and can be potentially published in Nature Comm. However, before further consideration, the following critical issues need to be addressed:

Authors' response: We appreciate the recognition of the value of this work by the reviewer. We have however revised some of the statements in this revision: 1) we used TALP (tungstate anion linked polyaniline) to replace POAFs. 2) we focused on the nanofluidic structure and tuned down the emphasis on the channel alignment. The reasons have been explained in our responses above.

-The highly aligned sub-2 nm pores with fast ion diffusion are the core concepts of the current work. However, there is little structural proof that can disclose the sub-2 nm hydrated nanofluidic channels based on the synthesized POAFs. Actually, the chemical nature and chemical descriptions of the key materials POAFs are poorly defined in the current work. It is really hard to understand what are the chemical structures of POAFs, and how the sheets are stacked in the solid-state through non-covalent interactions. From Fig 1 and 1b, it is also difficult to discern the interlayer structures of POAFs; therein, the authors shall provide high-resolution TEM images to reveal the interlayer and intralayer structures.

Authors' response: These are valuable suggestions. We have included more results in the main text to describe the structure of the TALP material (Fig 1). The chemical interaction that holds the TALP structure is hydrogen bond and it was a big challenge to collect high-resolution TEM to reveal the interlayer and intralayer structures. From the current low-mag TEM image below, the interlayer structure is visible when the image is enlarged sufficiently (left). Some resolution will be lost when it is compressed smaller (right).

We put together a descriptive structural model in Figure 1 in the Revision (and below), which is coherent with the conceptual model we described based on chemical analysis in previous publications (ref 15, 16). Briefly, the interlayer structure of TALP consists of dangling tungstate anions that are hydroscopic and have very thin layer of water molecules attached on them. The intralayer structure is constituted of polyaniline chains that are cross-linked by the tungstate anions, the interaction between which is hydrogen bond. The multiple parallel hydrogen bonds between the tungstate and the polyaniline are collectively strong to stabilize the structure.

The stacking of the nanosheets is driven by a collection of hydrogen bonding, electrostatic and ionic interactions. We think the hydrogen bonding could be more dominant because we observed interlayer swelling when the water molecules are replaced with less polar molecules.

The molecular structure of TALP has been discussed to some extent in our previous publications (ref 15,16). The understanding of the nanofluidic feature is the new finding in this work and is critical to the electrode design for gel supercapacitors.

-The swift ion transport through the POAF frameworks is crucial in the current work, but why sub-2 nm nanochannels can be the best? More understanding and proof of ion transport by comparison of different samples can be helpful. It seems that with high-pressure compression, it is possible to tune the nanochannels' sizes. Did the authors check other channel sizes dependent ion transport and electrochemical performance?

Authors' response: Whether sub-2nm or a particular size of the channels is the best is dependent on the specific electrolyte chemistry, including the ion size, solvation and solvent molecule size. When we discussed sub-2nm in the original version, it is suggested by many recent publications (ref 10, 12, 19, 20, 21) based on the collective experimental observation of a range of supercapacitor materials. The real science governing this phenomenon is not fully understood but it is probably relevant to the overlapping of the electric double layer (EDL) on the opposite surfaces of the channel. The overlapped EDL could produce unified directional electric field to speed up the ion transport. We understand the uncertainty in regards of the optimal size for ion transport and we realize the channel size is not dominant in the experimental findings reported in this work. As such, we refined our description and elaboration of the results to the nanofluidic structure. The terminology of sub-2nm channel is now merely used as descriptive data to understand the dimension/geometry of the nanofluidic channel and we avoided correlating the channel size with the performance.

With high pressure compression, or other mechanical approaches, the channel size can be tuned. We have obtained some interesting data on how the wider channels can make the ion conduction faster. Those results are submitted for review with another journal. We could consider sharing the draft if required.

As to ‘the comparison with other samples to prove the ion transport’, we compared the TALP with porous graphene in Figure 3e the Revision (and below). The results are very interesting and are discussed above. Here is a summary. In liquid electrolyte cell, where the electrodes are flooded by electrolyte, both TALP and graphene electrodes perform reasonably well. In gel electrolyte, where the electrodes are not flooded, the TALP performs similarly with its behaviour in liquid electrolyte whereas graphene’s performance drops drastically. We treat this contrast electrode behaviour as the clear sign to prove the function of nanofluidic ion channels for gel supercapacitor.

Minor comments: the calculation of knee frequency and time constants shall be provided in SI. The equations for the analysis of ion diffusion coefficient and activation energy shall be included in the main text.

Authors’ response: Thanks for the suggestion. We updated the calculation methods (Supplementary Fig 6) and the equations in main text (Page 9 and Page 11 in Revision).

Reviewer #3 (Remarks to the Author):

The authors demonstrated the conductive polymer-oxyanion frameworks (POAFs) possessing dense alignment of sub-2nm ion channels for rapid ion diffusion. When fabricated in thick electrodes, POAFs exhibit excellent areal and volumetric capacitance in solid-state aqueous electrolyte. While this manuscript needs a major revision before its consideration for publication. The comments are presented as follows.

Authors' response: Thanks for considering this work. As above statement, POAFs is replaced with TALP to make the description more accurate and meaningful.

(1) It is suggested to introduce the research background of POAFs. The novelty of this manuscript should be further explained when considering their published articles related to POAFs (Ref. 23 and 24)

Authors' response: Thanks for this comment. As discussed in the Revision, this work is not about the synthesis of the TALP material. We reported the synthesis of TALP in ref 15 and 16.

This work has the following breakthroughs: 1) combine the TEM and NMR analyses to discover the nanofluidic structure of TALP; 2) propose the use of nanofluidic structure to enhance the electrode capacitance in gel electrolyte to the level in liquid electrolyte; 3) demonstrate the impact of applying nanofluidic electrode on the gel supercapacitor performance.

Despite the use of the same materials as our previous publications, this manuscript reports the new, more profound understanding and more impactful application of the TALP materials for quasi-solid-state or solid state energy storage devices.

We understand the innovation in materials synthesis in this work is minimal. However, the innovation of new properties and new application potentials are magnificent in this work.

(2) In page 6-7, the authors claimed “the predominant orientation of nanosheets along the pellet axis” based on XRD and SEM analyses. It is better to explain why the POAFs nanosheets are selectively orientated solely upon compaction. Is there something like inductive effect?

Authors' response: Thanks for this comment. As we discussed above, the alignment is bi-directional and is more inclined to the vertical direction. This process

is not controllable in our current pressing setup. We think extra mechanical experiments are necessary to fully understand the lubricating and sliding behaviour of the TALP particles, which should well deserve an independent study. To prevent misunderstanding, we avoided correlating the alignment with the electrode performance. We might revisit this plausible effect if we can control the alignment of the channels.

(3) The comparison of capacitance performance of this work with others lacks rigor (Fig. 3f, 4a). For example, Ref 10, the electrodes were measured in IL electrolyte, different from this work using aqueous electrolyte. As we known, IL electrolyte gives lower capacitance but significantly wider potential window. In addition, state-of-the-art pseudocapacitive examples should be included for comparison.

Authors' response: Thanks for this comment. In our original submission, we only compared the capacitance of gel-based electrode. As we discussed above, there are very limited number of publications that report the areal, gravimetric and volumetric capacitances and the ref 10 (a Nature Energy paper) is one of these. Moreover, the gravimetric and volumetric capacitances in this ref 10 are superior to many other even in non-IL gel electrolytes (Fig 4a). When we compare, we do not target on a single paper, but on a group of porous electrodes. This ref 10 is one typical example of graphene/graphene oxide laminar composite with tuneable pore size. On account of these thoughts, omitting this ref 10 seems inappropriate either.

To avoid misunderstanding, we have added a short explaining sentence in the figure captions:

Fig 3&4: *Note that ionogel and ionic liquid electrolytes used in reference 10.*

We also compared the Fig 4a based on Charge (see above response to Reviewer 1 and Supplementary Fig 11).

As to the pseudocapacitive examples, we didn't find sufficient data in related works that allow us to calculate the three capacitance metrics in gel supercapacitors. Therefore, we are unable to put those in the comparison. The compared example includes only ref. 9 (MXene) in Fig 4a.

Lastly, performance comparison in our work is not purposed to reporting record high value. We focus on how the nanofluidic electrode design could introduce value-adding advances to the gel supercapacitors. The difference in capacitance is the experimental evidence that support our theory. Our findings could be interpreted in more meaningful ways: 1) nanofluidic TALP electrode can balance areal, gravimetric and volumetric capacitances without obvious constraint on one single metric; 2) nanofluidic TALP electrode can reduce the use of infilling gel and hence reduce the total volume and mass of the device; 3) nanofluidic TALP can maintain comparable capacitance in gel to that in liquid.

(4) The specific surface area and pore size distribution of POAFs should be provided.

Authors' response: We have added the specific surface area data ($\sim 22 \text{ m}^2/\text{g}$). Due to the very low porosity, the N_2 cryo-sorption analysis does not derive the pore size distribution. We have used microCT to measure the total porosity of the compacted TALP pellet (Fig 2).

(5) The thickness values of compacted POAFs pellets with different mass loadings should be provided.

Authors' response: These data are added in Supplementary Table 3.

	Mass loading (m g cm^{-2})	Thickness (μm)	Areal capacitance (F cm^{-2})		
			1 m A cm^{-2}	10 m A cm^{-2}	30 m A cm^{-2}
Liquid electrolyte	10	56	1.80	1.28	0.89
	20	111	3.45	2.35	1.61
	30	167	5.25	3.57	2.39
	40	222	6.99	4.50	2.89
	50	278	9.10	6.28	3.92
Gel electrolyte	10	56	1.84	1.27	0.72
	20	111	3.59	2.36	1.40
	30	167	5.38	3.29	1.88
	40	222	7.23	4.25	2.37
	50	278	8.94	5.30	3.06

Reviewers' Comments:

Reviewer #1:

Remarks to the Author:

The authors performed a thorough revision of the manuscript. While I still have concerns regarding the low charge-discharge rate and application of a linear equation to nonlinear galvanostatic discharge curves, I feel that the manuscript has enough novelty and contains sufficient basic science to justify publication.

Reviewer #2:

Remarks to the Author:

In the revised manuscript, the authors have made additional efforts to address the issues raised by the reviewers. Most of my concerns have been resolved, except for the quality of the TEM image demonstrating the interlayer/intralayer nanofluid structures of TALP. The TEM image presented is not of high enough quality in my opinion. Nevertheless, I leave it to the editor to decide if additional effort is needed.

Minor comments:

-It would be helpful if the authors could include the values of the achieved areal/volumetric/gravimetric capacity in the abstract.

-Since the authors can control the size of the nanofluid channels and achieve even better capacitance performance, it would be good if the authors include this perspective/statement in the conclusion section.

Reviewer #3:

Remarks to the Author:

Notable changes have been made in this revised manuscript. Considering that the electrode materials have been reported in their previous reports, here the authors modified their focus on the nanofluidic structure of electrode for enhanced capacitance in gel electrolytes. It is the case that the TALP electrode in Na₂SO₄-based gel electrolyte exhibits a good balance of areal, gravimetric and volumetric capacitances even at very high mass loadings. My major reserved comment is the inappropriate comparison of capacitances of TALP electrode with other porous electrodes. Since the TALP is a hybrid of tungstate and polyaniline with a low specific surface area ($\sim 22 \text{ m}^2/\text{g}$), the capacitance of TALP electrode should be dominated by pseudocapacitance but not EDL capacitance. However, except for reference 9, the compared references (5-8, 10, 12, 31, 38, and 39 in Fig. 3e, 3h, and 4) utilized EDLC-type materials (graphene or activated carbon). It is misleading to compare the capacitance of two-type electrode materials of different storage mechanisms, especially under the condition that the power and cycling performances of TALP electrode are inferior to the EDLC-type materials. Moreover, as mentioned in my previous version, comparison of electrode materials in aqueous and non-aqueous electrolytes also lacks rigor. The authors attribute the inappropriate comparison to "the very limited number of comparable references in gel supercapacitors". In my opinion, it is not necessary to "choose the publications on gel supercapacitors that have provided all of the three capacitance metrics". Otherwise, it is better to delete the related comparative figures rather than displaying them in the main text.

As minor comment, I agree with Review 1's comments to show energy/power densities, and to calculate the capacitance by integration.

Responses to REVIEWER COMMENTS

Reviewer #1 (Remarks to the Author):

The authors performed a thorough revision of the manuscript. While I still have concerns regarding the low charge-discharge rate and application of a linear equation to nonlinear galvanostatic discharge curves, I feel that the manuscript has enough novelty and contains sufficient basic science to justify publication.

Authors' response: We appreciate the positive comments.

We understand the concerns of the reviewer on the charging rate, which is regarded a general challenge for solid state supercapacitors and batteries. Solving the high rate chargeability of solid state devices will require efforts from both electrode and electrolyte improvement, without compromising the trade-off relationships among high mass loading, large density and small volume of electrode. We will endeavour to solve this challenge by taking advantages of nanofluidic electrode design and electrolyte modulation in our further studies.

We also acknowledge the reviewer's concern on the use of linear equation for nonlinear galvanostatic charge discharge (GCD) curves. In our response to Reviewer #3, it is shown the Mathias integration equation could be surprisingly transformed to the linear equation. Although there are elusive scientific questions in Mathias equation (discussed in our response to Review #1 in the First Revision), the idea of avoiding linear equation in non-linear GCD calculation is fair. At this moment, unfortunately, we don't think there is a reliable and peer-reviewed equation to satisfy the reviewer's comment.

Reviewer #2 (Remarks to the Author):

In the revised manuscript, the authors have made additional efforts to address the issues raised by the reviewers. Most of my concerns have been resolved, except for the quality of the TEM image demonstrating the interlayer/intralayer nanofluid structures of TALP. The TEM image presented is not of high enough quality in my opinion. Nevertheless, I leave it to the editor to decide if additional effort is needed.

Authors' response: We appreciate the recognition of the value of this work by the reviewer.

Because of the beam sensitiveness of TALP, we currently are unable to acquire the atomic resolution TEM image of the intralayer structure. Daliang Zhang, Kun Li, and Yu Han's team published a Science paper in 2018 (Atomic-resolution transmission electron microscopy of electron beam-sensitive crystalline materials), which reported a suite of methods to overcome the challenges in soft materials TEM. This will be a direction of future studies on TALP.

For the moment, we have endeavoured to acquire higher resolution TEM image of the interlayer stacking with as shown below. Note the high-quality electron diffraction pattern indicates the order of interlayer stacking in TALP.

Fig. 2b The side-view TEM image of TALP.

Minor comments

-It would be helpful if the authors could include the values of the achieved areal/volumetric/gravimetric capacity in the abstract.

Authors' response: We appreciate this suggestion. We have revised the abstract accordingly.

Revision:

Taking advantages of the intrinsic nanofluidic channels of voidless electrodes, we demonstrate exceptional capacitance enhancement in gel electrolyte with negligible loss (~1.8%) relative to that in liquid electrolyte. In gel electrolyte, the areal capacitance reaches 8.94 F cm^{-2} with a gravimetric capacitance of 178.8 F g^{-1} and a volumetric capacitance of 321.8 F cm^{-3} .

-Since the authors can control the size of the nanofluid channels and achieve even better capacitance performance, it would be good if the authors include this perspective/statement in the conclusion section.

Authors' response: Thank you for this suggestion. We have included relative comments in the conclusion section.

Revision:

Adjustment of the channel size will promisingly extend its use to gel electrolytes based on organic solvents or ionic liquids.

Reviewer #3 (Remarks to the Author):

Notable changes have been made in this revised manuscript. Considering that the electrode materials have been reported in their previous reports, here the authors modified their focus on the nanofluidic structure of electrode for enhanced capacitance in gel electrolytes. It is the case that the TALP electrode in Na₂SO₄-based gel electrolyte exhibits a good balance of areal, gravimetric and volumetric capacitances even at very high mass loadings.

Authors' response: Thanks for this comment. As the reviewer remarked, this work for the first time demonstrates how nanofluidic channels can make gel-based supercapacitors to achieve balanced capacitance performance on universal metrics. This breakthrough has solid scientific value and will show practical technical impact.

My major reserved comment is the inappropriate comparison of capacitances of TALP electrode with other porous electrodes. Since the TALP is a hybrid of tungstate and polyaniline with a low specific surface area (~22 m²/g), the capacitance of TALP electrode should be dominated by pseudocapacitance but not EDL capacitance.

Authors' response: Thanks for this comment. As the reviewer commented, this TALP electrode has very low surface area and porosity, which is the unique advantage to capacitance enhancement in gel electrolyte.

However, in our opinion, it is inappropriate to justify the type of capacitance based on surface area. The specific surface area (SSA) is measured by Nitrogen cryosorption under dry and low-pressure conditions. In electrode measurement, there is a term named Electrochemical Accessible/Active Surface Area (EASA). EASA is usually not measurable in electrolyte but can be estimated based on the value of capacitance. EASA for meso/macroporous carbons is often consistent with the SSA but could be significantly less than SSA for microporous carbons because of the slow ion diffusion and low electrolyte wettability. The nanofluidic channels of TALP are highly hydrated that enable the excellent ion accessibility, which means its EASA is larger than SSA. This scenario is not the same as traditional porous carbons. In fact, this point presents the spirit of this work – to re-invent capacitive materials from a non-porous perspective. We hope the above explanation would make sense to the reviewer.

On the type of capacitance, although polyaniline is a well-known pseudocapacitive material, its pseudo capacitance is largely from protonation reaction in acid at

specific electrode potentials. In this work, we used neutral Na₂SO₄ electrolyte instead of acidic electrolyte. To differentiate the Faradaic behaviour of TALP in acidic electrolyte, we have compared the cyclic voltammetry curves as below. For TALP, the pseudo (redox) capacitance is dominant in acidic electrolyte. In contrast, seriously suppressed redox capacitance is observed in neutral electrolyte, which is reasonably attributed to EDL-like non-faradaic surface processes.

However, except for reference 9, the compared references (5-8, 10, 12 31, 38, and 39 in Fig. 3e, 3h, and 4) utilized EDLC-type materials (graphene or activated carbon). It is misleading to compare the capacitance of two-type electrode materials of different storage mechanisms, especially under the condition that the power and cycling performances of TALP electrode are inferior to the EDLC-type materials.

Authors' response: In fact, carbon materials, especially graphene (normally produced by reduction of graphene oxide) and activated carbons (produced by activation of polymers or biomass precursors), contains oxygen functional groups that contribute pseudocapacitance in addition to the EDL capacitance. As listed in Supplementary Table 1, the carbon electrodes compared in Fig 4a used either H₂SO₄ or KOH electrolytes. There are enormous studies on the pseudocapacitance of carbon materials in the acidic and alkaline electrolytes, which is oftentimes related to the surface functional groups of carbons. In this context, it is difficult to argue the type of capacitance without strict definition of the material properties and the electrochemical cell conditions (electrolyte, electrode potential, etc.).

The comparison in Fig 3e is to illustrate the capacitance difference between gel and liquid electrolytes. The comparison is to reveal the fact that porous electrode (carbon is the most typical porous materials in electrochemical capacitor) cannot store charges without gel infiltration. The capacitance of graphene and TALP in liquid electrolyte is not the point of comparison. The reviewer is suggested to double check the relevant discussion on the comparison in Fig 3e in our manuscript and we do not mention anything about the low capacitance of graphene in liquid electrolyte – comparing the value of capacitance is not the scientific focus of this work. We hope these comments will help elucidate the meaning of the work to the reviewer.

In the Second revision, Fig 3h is replaced with a comparison figure of volumetric capacitance. The areal capacitance in the original Fig 3h is displayed in Supplementary Fig 10. The comparison in both Fig 3h and Suppl. Fig10 is to show the relationship between areal/volumetric capacitance and electrode thickness for TALP and porous electrodes. Theoretically increasing electrode thickness should linearly increase areal capacitance without impairing volumetric capacitance. But the accumulated ion resistance in thick electrode will reduce the material utilization efficiency and lead to the loss of capacitance. We note this analysis with a single sentence comment: *By comparison, some porous electrodes usually demonstrate gradual loss of volumetric capacitance as the thickness (mass loading) increases.* We would stress that the relationship in Fig 3h and Suppl. Fig 11 is independent of the type of capacitance (related to surface electron transfer) but is essentially determined by the mass transport kinetics of electrode structure.

Fig 4a is a very important comparison figure. The key message of Fig 4 is to inspire researchers in this field to re-evaluate the way of performance benchmarking. As noted previously, most of the studies only compare one of the gravimetric, volumetric, areal capacitances (some does compare two of the three). From technical reliability perspective, we believe it is necessary to fairly compare all three capacitance metrics. This is very important for the scientific advancement of the field as we have seen many works solely focus on reporting high value of performance with low mass loading, thin electrode or highly porous skeleton, which will introduce practical challenges for applications. As the reviewer acknowledged, in Fig 4a, we intended to demonstrate the feasibility of the nanofluidic electrode to keep appropriate balance among areal, volumetric and gravimetric capacitances.

Fig 4b discloses another overlooked factor in gel supercapacitor design – the mass of gel electrolyte that is taken up by electrode. In battery manufacturing, the amount of electrolyte is always strictly controlled to reduce the weight of entire device. Our work shows the opportunity of reducing electrolyte amount in supercapacitor therefore paving the way to improve overall device capacitance.

In general, we felt that the Reviewer #3 assumed the higher capacitance of TALP is concluded from the comparison by tentatively ignoring other data that should be included in the opinion of the Reviewer #3. As we have explained, we do not focus on capacitance comparison but on the impact of electrode on ion accessibility in gel electrolyte. Capacitance is used as a measurable factor to reflect ion accessibility. We hope our rationale is justified to Reviewer #3. Again, we wish to explain that this work is not to report the high capacitance of TALP relative to some carbon materials, but to demonstrate the potential of nanofluidic electrode structure as a promising alternative electrode to carbon materials. Last but not least, we will be very grateful if the Reviewer #3 could recommend relevant justifiable works to be included in the comparisons.

Moreover, as mentioned in my previous version, comparison of electrode materials in aqueous and non-aqueous electrolytes also lacks rigor.

Authors' response: We have removed the data of organic gel electrolyte in the comparison in Fig 4a, without impairing the conclusion.

The authors attribute the inappropriate comparison to “the very limited number of comparable references in gel supercapacitors”.

Authors' response: We would claim that the lack of reports on all three areal/volumetric/gravimetric capacitance metrics is the current situation of the field. Not reporting all three metrics is nothing wrong, but it will enable more scientific insights if the full spectrum of the capacitances is reported.

In my opinion, it is not necessary to “choose the publications on gel supercapacitors that have provided all of the three capacitance metrics”.

Authors' response: As explained above, we think it will create long term benefit in the community by fairly reporting the three capacitance metrics. In terms of practical

usage, the volume, mass and footprint area are all essential factors to consider for device design. We strongly believe the need to keep Fig 4a.

As the Reviewer implicated, we could easily find many works that report solely on areal, volumetric or gravimetric capacitance. But what is the scientific meaning and technical impact of doing so? We will be appreciated if Reviewer 3 could clarify this issue.

We study the gel electrolyte supercapacitors because of the challenge of infiltrating gel into the porous electrode. The value of our work can be justified by referencing the Nature Communication publication (ref. 4, *High energy flexible supercapacitors formed via bottom-up infilling of gel electrolytes into thick porous electrodes. Nature Communications* **9**, 2578), which reported a method to improve the gel infiltration efficiency into thick porous electrode. Our work offers a conceptually unique method to get rid of the gel infilling procedure and reveals the feasibility of applying nanofluidic mechanism to thick electrode design in gel electrolyte.

Based on the above discussion, we believe it is necessary and meaningful to report the three capacitance metrics in gel supercapacitors.

Otherwise, it is better to delete the related comparative figures rather than displaying them in the main text.

Authors' response: We have clearly expressed our opinions on the foundation and importance of the comparisons in the above responses. We want to clarify that this work is not to disclaim the feasibility of carbon materials for supercapacitors, but to provide a different angle to the design principle of capacitive electrode. These comparisons are critical to demonstrating the feasibility of nanofluidic electrode. Unfortunately, we cannot agree with the reviewer to remove all these comparisons. Nonetheless, we are open to improve the comparison if the reviewer will provide clear suggestions, e.g. inclusion of other suitable porous materials.

As minor comment, I agree with Review 1's comments to show energy/power densities, and to calculate the capacitance by integration.

Authors' response: Thanks for this comment. We have addressed these in detail in our response to Reviewer 1 in the First Revision. It seemed we have convinced Reviewer 1 that this work is focused on capacitance enhancement, and we do have

other works that can increase the working voltage. Therefore, it is not necessary to calculate the energy/power density and we have reported all relevant data so readers can easily work out the energy/power densities on their own. To prevent misleading understanding, we have revised the title again to '*Nanofluidic voidless electrode for electrochemical capacitance enhancement in gel electrolyte*'. We believe this change has clearly defined the nature of the study.

There are elusive scientific questions of applying Mathias equation and we have discussed these matters with Reviewer 1 in the First Revision. We also discussed these issues with Mathias who acknowledged this issue and indicated his plan to discuss with his colleagues (since then we haven't heard from Mathias). We also found that the Mathias integration equation was convertible to the linear equation with the V(t) as the entire voltage window, as shown below.

$$C = I \times \int \left(\frac{1}{V(t)} \right) dt = I \times \frac{1}{V(t)} \times \int dt = \frac{I \times t}{V(t)}$$

Reviewers' Comments:

Reviewer #3:

Remarks to the Author:

Two reserved comments are the same as my previous version.

1. I still doubt the storage mechanism of TALP, a hybrid of tungstate and polyaniline, in the neutral aqueous electrolyte. In the authors' Responses, they attributed the storage mechanism of TALP to the EDL-like non-faradaic surface processes according to the suppressed redox capacitance in neutral electrolyte than in acidic electrolyte. It is insufficient to determine a charge mechanism solely based on electrochemical performance. In addition, the faradaic charge storage processes of polyaniline were also reported in neutral aqueous electrolytes (Adv. Funct. Mater. 2018, 28, 1804975; Electrochimica Acta 2018, 268, 131). Therefore, it is better if typical pseudocapacitive materials (such as Mxene, organics, MOFs/carbon composite, not only functional carbons) are included for comparison.

2. For the integration equation of capacitance calculation:

$$C = I \times \int [1/V(t)] dt$$

the $V(t)$ corresponds to the potential change during the time change dt . This integration equation can be converted to the linear equation ($I \times t/V$) for the linear GCD curves where the $V(t)$ is the same at a given dt . However, for the non-linear GCD curves as like those in this work, $V(t)$ is a variable and the conversion of integration equation to linear equation is not correct. It is more accurate to calculate capacitance by integration.

Overall, the focus of this manuscript is the nanofluidic electrode for well-balanced gravimetric/volumetric/areal capacitance in gel electrolyte. Although the minor problems mentioned above doesn't significantly affect the scientific importance of the work, the authors are suggested carefully revise them.

Responses to REVIEWER COMMENTS

Reviewer #3 (Remarks to the Author):

Two reserved comments are the same as my previous version.

I still doubt the storage mechanism of TALP, a hybrid of tungstate and polyaniline, in the neutral aqueous electrolyte. In the authors' Responses, they attributed the storage mechanism of TALP to the EDL-like non-faradaic surface processes according to the suppressed redox capacitance in neutral electrolyte than in acidic electrolyte. It is insufficient to determine a charge mechanism solely based on electrochemical performance. In addition, the faradaic charge storage processes of polyaniline were also reported in neutral aqueous electrolytes (Adv. Funct. Mater. 2018, 28, 1804975; Electrochimica Acta 2018, 268, 131). Therefore, it is better if typical pseudocapacitive materials (such as Mxene, organics, MOFs/carbon composite, not only functional carbons) are included for comparison.

Authors' response: We appreciate the useful comments. We have included additional data as specified in Supporting Table 1, which include MXene/MOF (ref 9), polyaniline (ref. 40) and graphene oxide (ref. 41).

As it is not the focus of this work, we did not investigate the chemical state change of TALP during charge and discharge. In our previous report (Adv. Mater. 2018, 30, 1800400), we used XPS to track the surface state of polyaniline chains in TALP during charging and discharging, and did not determine measurable chemical shift, which is strong evidence to indicate the EDL capacitance is the core contribution. Certainly, we won't deny the pseudocapacitance contribution – but it is a matter of the percentage.

For the integration equation of capacitance calculation:

$$C = I \times \int [1/V(t)]dt$$

the $V(t)$ corresponds to the potential change during the time change dt . This integration equation can be converted to the linear equation ($I \times t/V$) for the linear GCD curves where the $V(t)$ is the same at a given dt . However, for the non-linear GCD curves as like those in this work, $V(t)$ is a variable and the conversion of integration equation to linear equation is not correct. It is more accurate to calculate capacitance by integration.

Authors' response: Thank you for this suggestion.

1) We have independently developed a new integral equation to calculate the capacitance from non-linear GCD profile, which shows negligible deviation from the linear equation. Detailed discussion has been added in the Main text Methods and Supporting Information.

2) We are afraid that the equation ($C = I \times \int [1/V(t)]dt$) in 10.1002/aenm.201902007 Adv Energy Mater paper (Mathis paper) is not correct, although this paper does point out a key issue on the capacitance calculation accuracy.

As the 3rd reviewer considered, if $V(t)$ is the voltage change for a time interval of dt , then $I \times dt/V(t) = dC(t)$. Thus, the Mathis equation can be converted to the integration of $dC(t)$ on time: $C = \int dC(t)$, which means, by dimension, F (capacitance) equals to F x s (capacitance x time). The physical meaning of this equation is incorrect.

2) As Prof Yury Gogotsi has reported in his Nature article (Conductive two-dimensional titanium carbide 'clay' with high volumetric capacitance, 2014, 516, 78–81), the integration equation is used when the capacitance is measured by cyclic voltammetry.

$$C = \frac{1}{\Delta V} \int \frac{j dV}{s}$$

A linear equation is used when the capacitance is measured by galvanostatic charge-discharge. Please note that MXene is a typical pseudocapacitive material, which also presents non-linear GCD curve (Fig 3b in Nature Energy 2019, 4, 241–248, by Yury et al.)

$$C = (jt) / V$$

We also found many Nature Commun papers used this linear equation to calculate GCD capacitance of non-linear GCD curves. Below is a list of these articles.

Article	Material	GCD capacitance calculation	GCD curve linearity
Nat Commun 12, 4297 (2021)	polypyrrole	$C = \frac{I \times \Delta t}{S \times \Delta U},$ ΔU is the voltage window	Non-linear (Fig 5 b & e)

		S is electrode area	
Nat Commun 6, 7260 (2015).	Ni/reduced graphene oxide	$C_V = I \times t \times U^{-1} \times V^{-1}$ U is the voltage window, V is the electrode volume	Non-linear (Fig 5c & 7)
Nat Commun 8, 536 (2017).	MnO	Specific capacitance (C) = $\frac{I\Delta t}{\Delta V}$. (for galvanostatic charge/discharge measurement)	Non-linear (Fig 5b)
Nat Commun 5, 3754 (2014).	Reduced graphene oxide	$CA = I \times t \times U^{-1} \times S^{-1}$ U is the voltage window S is electrode area	Non-linear (Fig 3g-l, 4a, 4c, 5e, 6b)
Nat Commun 11, 1843 (2020).	Active carbon	$C_S = \frac{4It}{mV}$	Non-linear (Fig 4b, 5h)
Nat Commun 11, 4712 (2020).	Covalent organic framework	$C_g = 4I\Delta t/m\Delta V$	Non-linear (Fig 4b)
Nat Commun 10, 4913 (2019).	MnO ₂	$C = \frac{It}{\Delta V}$,	Non-linear (Fig 6)
Nat Commun 11, 3884 (2020).	Functionalized carbon	$C_S = \frac{I \times \Delta t}{m \times \Delta V}$	Non-linear (Fig 5b)

We would note that the reference (Electrochimica Acta 2018, 268, 131) suggested by the reviewer used the linear equation ($C=Ixt/V$) to calculate the capacitance of pseudocapacitive polyaniline.

Overall, the focus of this manuscript is the nanofluidic electrode for well-balanced gravimetric/volumetric/areal capacitance in gel electrolyte. Although the minor problems mentioned above doesn't significantly affect the scientific importance of the work, the authors are suggested carefully revise them.

Authors' response: We appreciate the positive comments.